# Enzymatic protein fusions with 100% product yield

**Adrian CD Fuchs***

Department of Protein Evolution, Max Planck Institute for Biology, Tübingen, Germany

## eLife Assessment

This revision of **important** work is a versatile addition to the chemical protein modifications and bioconjugation toolbox in synthetic biology. The technology developed cleverly uses Connectase to irreversibly fuse proteins of interest together so they can be studied in their native context, with **compelling** well-controlled data showing the technique works for various protein partners. This work will help multiple fields to explore multi-function constructs in basic synthetic biology. This work will also be of interest to those studying fusion oncoproteins commonly expressed in various human pathologies.

**Abstract** The protein ligase Connectase can be used to fuse proteins to small molecules, solid carriers, or other proteins. Compared to other protein ligases, it offers greater substrate specificity, higher catalytic efficiency, and catalyzes no side reactions. However, its reaction is reversible, resulting in only 50% fusion product from two equally abundant educts. Here, we present a simple method to reliably obtain 100% fusion product in 1:1 conjugation reactions. This method is efficient for protein-protein or protein-peptide fusions at the N- or C-termini. It enables the generation of defined and completely labeled antibody conjugates with one fusion partner on each chain. The reaction requires short incubation times with small amounts of enzyme and is effective even at low substrate concentrations and at low temperatures. With these characteristics, it presents a valuable new tool for bioengineering.

*For correspondence:
adrian.fuchs@tuebingen.mpg.de

## Introduction

Protein conjugations are used to fuse fluorophores to proteins and to immobilize proteins on beads, microplates, electron microscopy grids, or surface plasmon resonance chips. They enable the generation of antibody conjugates, of segmentally labeled proteins for nuclear magnetic resonance measurements, and the fusion of proteins to the cell surface. Various methods are employed for these applications, often with an excess of one conjugation partner (e.g. 3–10 equivalents). Each of them comes with its own advantages and disadvantages. N-Hydroxysuccinimide (NHS) labeling of lysine residues or maleimide labeling of cysteine residues, are easy to use, but can be unspecific (*Stephanopoulos and Francis, 2011*). Click chemistry is specific, but requires the introduction of non-biological chemical groups in the fusion partners (*Kaur et al., 2021*; *Kolb et al., 2001*). Split domain methods, such as SpyTag-SpyCatcher, are also specific, but introduce long sequences between the fusion partners (*Zakeri et al., 2012*; *Sutherland et al., 2019*). Split intein methods introduce only a short 'ligation scar', but may suffer from solubility problems and varying efficiency for different substrate pairs (*Wu et al., 1998*; *Anastassov et al., 2024*). Enzymatic methods are simple and, in some cases, specific, but the resulting fusions are reversible and therefore incomplete (*Schmidt et al.,*

*2017*; *Morgan et al., 2022*). Nonetheless, they find good use, both in industry and in thousands of academic studies.

The most popular enzyme, *Staphylococcus aureus* Sortase A, catalyzes the reversible fusion of substrates **A** and **B** in the form of **A**-LPXTG (X=any amino acid) and $G_{1-5}$-**B** to yield **A**-LPXTG$_{1-5}$-**B** (usually with additional linker sequences) (*Morgan et al., 2022*; *Mazmanian et al., 1999*). Therefore, it shows good specificity for substrate **A**, but is relatively unspecific towards substrate **B**. Here, the glycine may be replaced by lysine side chains or by other amines (*Heck et al., 2014*). Sortase A has a low affinity for its substrates and therefore requires high substrate concentrations for moderate activity e.g., $K_{M(LPETG)}$=7330 μM, $K_{M(GGGGG)}$=196 μM (*Frankel et al., 2005*; see also *Jacobitz et al., 2017*). It also catalyzes the irreversible hydrolysis of both educts and products ($k_{cat, \text{hydrolysis}}$ = 0.086 /s; $k_{cat, \text{Ligation}}$ = 0.28 /s) (*Frankel et al., 2005*). Some of these characteristics could be improved by the generation of mutant proteins or by the optimization of reaction conditions (*Chen et al., 2011*; *Chen et al., 2016*). Nevertheless, these properties still constrain its applicability, meaning that Sortase A is primarily used for protein-peptide fusions at high concentrations, and with an excess of peptide.

Several other enzymes share these basic characteristics with Sortase A. These enzymes belong to the cysteine (Sortase, Butelase, Asparaginyl endopeptidase) or serine proteases (Trypsiligase, Subtiligase), and bind their substrates through a (thio-)ester bond, which can react with $H_2O$ (hydrolysis, irreversible) or the $H_2N$ group of fusion partners (aminolysis, reversible) (*Schmidt et al., 2017*; *Morgan et al., 2022*). Recently, however, an enzyme ligase with entirely different characteristics was discovered (*Fuchs et al., 2021*). This enzyme, Connectase, binds its substrate through an amide bond, which is hydrolysis-resistant. It therefore exclusively catalyzes conjugations with the $H_2N$ groups of fusion partners. In addition, Connectase acts on a longer recognition sequence, leading to higher substrate specificity and higher catalytic efficiency. This enables entirely different applications, such as the specific labeling of proteins with fluorophores within cell extracts, allowing their in-gel detection with significantly higher sensitivity and signal-to-noise ratio compared to western blots with tag-specific antibodies (*Fuchs, 2023*).

Yet just like other enzyme ligases, Connectase catalyzes a reversible reaction (*Fuchs et al., 2021*). Connectase from *M. mazei*, for example, binds substrates with the sequence **A**-ELASKDPGAFDAD-PLVVEI (*Figure 1*, step 1). It then forms a covalent intermediate, **A**-ELASKD-Connectase, with the N-terminal part of this sequence, and cleaves off the C-terminal peptide PGAFDADPLVVEI (*Figure 1*, steps 2–3). This reaction works both ways, meaning that PGAFDADPLVVEI can react with **A**-ELASKD-Connectase to restore Connectase and its substrate, **A**-ELASKDPGAFDADPLVVEI. However, when a second substrate **B** in the form of PGAFDADPLVVEI-**B** is added to the reaction, it can be used instead of the peptide PGAFDADPLVVEI to form the fusion product **A**-ELASKDPGAFDADPLVVEI-**B** (*Figure 1*, steps 4–5).

When using equimolar quantities of educts **A** and **B**, Connectase catalyzes an equilibrium of approximately 50% fusion product **A-B** and 50% educts. This is because the same amounts of

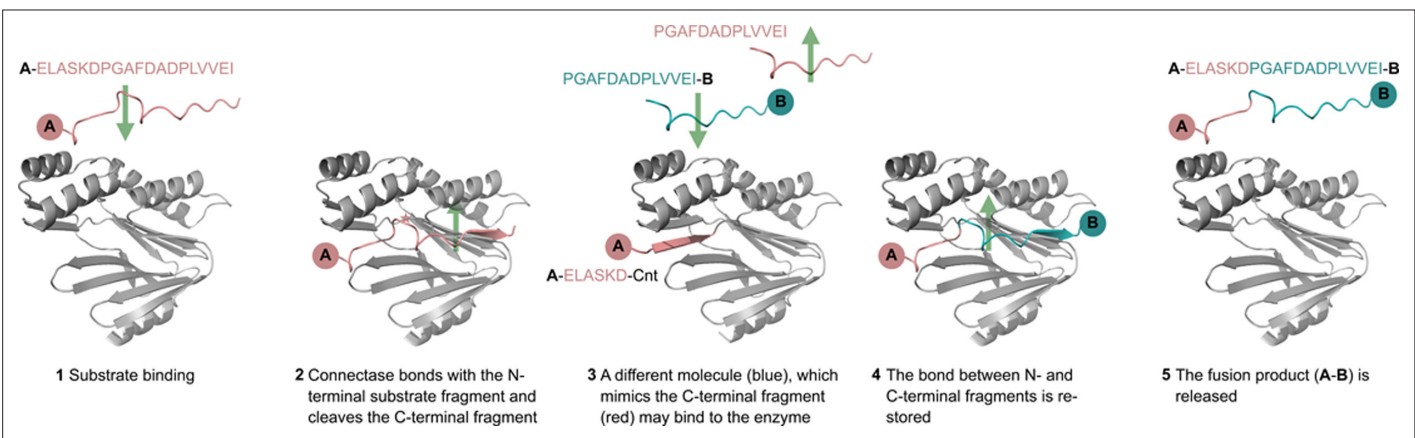

**Figure 1.** The Connectase reaction mechanism. The structures were predicted with Alphafold 2. For simplicity, they are shown mirror-inverted, so that the Connectase binding channel is visible, and the recognition sequence can be read from left (N-terminus) to right (C-terminus). A and B symbolize peptides or proteins.

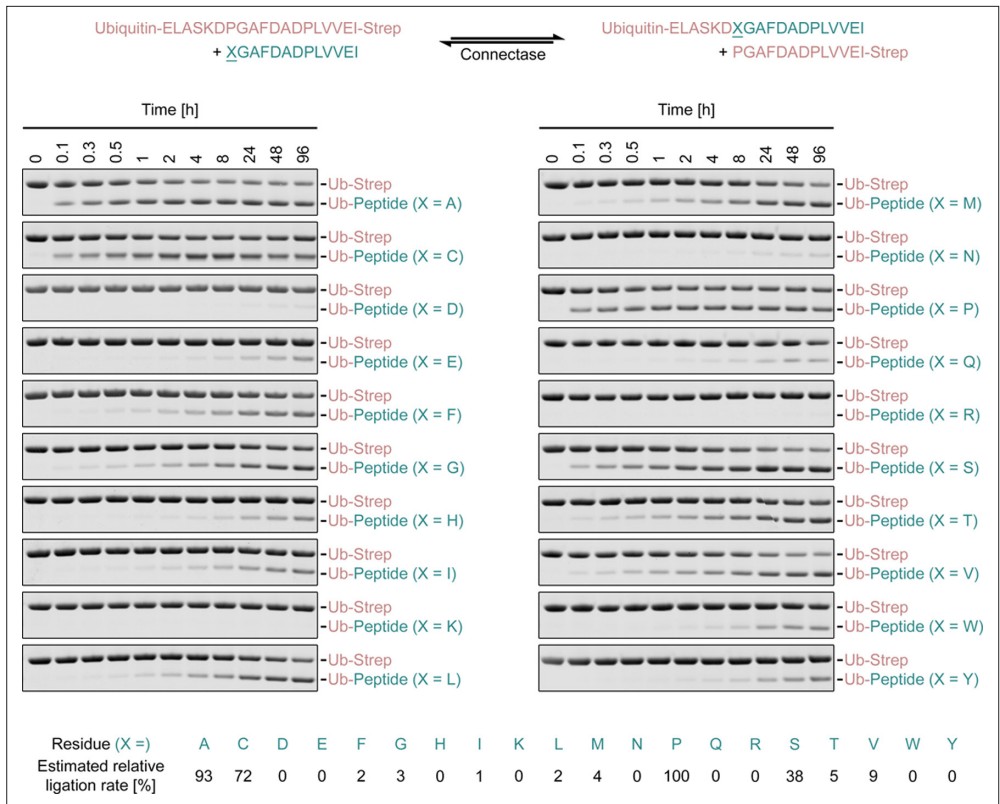

**Figure 2.** Mutagenic analysis of the Connectase recognition sequence. A ubiquitin substrate with a C-terminal Connectase recognition sequence followed by a Streptavidin-tag (Ub-Strep) is fused to peptides consisting of N-terminal Connectase recognition sequence variants (top scheme). These peptides differ in the first amino acid, where proline was replaced by each of the 19 other standard proteinogenic amino acids. An SDS-PAGE time course analysis of each reaction (1 eq. Ub-Strep, 1 eq. Peptide, 0.01 eq. Connectase, 22°C) shows the gradual emergence of the fusion product, Ub-Peptide. Based on densitometric analyses, the reaction rate with the different substrates was estimated (lower panel; an exact determination is not possible due to the reversibility of the reaction [see Methods section]) and normalized to the highest rate (X = P, 100%). The peptide substrates (XGADADPLVVEI) and byproducts (PGAFDADPLVVEI-Strep; top panel) are not visible on the gels.

The online version of this article includes the following source data for figure 2:

**Source data 1.** PDF file containing original gel images, indicating relevant bands.

**Source data 2.** Original gel images.

**Source data 3.** Measurements and calculations.

PGAFDADPLVVEI peptide byproduct and PGAFDADPLVVEI-**B** educt compete for the **A**-ELASKD-Connectase intermediate (*Figure 1*, step 3). Consequently, 100% fusion product can be obtained by removing or inactivating the peptide byproduct, for example by specific enzymatic proteolysis. However, the employed protease would have to act only on the PGAFDADPLVVEI peptide and not on the PGAFDADPLVVEI-**B** educt. In the following paper, we present a solution to this problem and describe a simple method to obtain 100% fusion product from equally concentrated educts in short time and with small amounts of enzyme.

## Results
### Modification of the Connectase recognition sequence

To facilitate the removal of peptide byproduct from Connectase reactions (see Introduction), we studied ways to alter the Connectase recognition sequence. For this, we used Ubiquitin (Ub) with a C-terminal Connectase recognition sequence followed by a Streptavidin tag (*Figure 2*, Ub-Strep). As a second reaction substrate, we used peptides derived from the Connectase recognition sequence

(XGAFDADPLVVEI, where X represents any of the 20 amino acids). The conjugation of these substrates results in a shorter Ubiquitin product (*Figure 2*, Ub-Peptide), which lacks the Streptavidin tag. Therefore, the conjugation rate with the different peptides could be determined by monitoring Ub-Peptide formation in SDS-PAGE time course analyses (*Figure 2*).

The experiment with the original recognition sequence peptide (X=Proline) resulted in the rapid formation of an equilibrium with ~0.5 equivalents (eq.) Ub-Strep and ~0.5 eq. Ub-Peptide, corresponding to ~50% product yield from equally abundant educts (1 eq. each; see introduction). While proline substitutions with X=D, E, F, G, H, I, K, L, M, N, R, T, W, or Y drastically reduced ligation rates, substitutions with S, C, and A resulted in moderate or high ligation rates. This was surprising, because the KDPGA sequence is highly conserved in the physiological Connectase target, mtrA (methyltransferase A) (*Fuchs et al., 2021*). Due to its unique structure and chemistry, proline was previously considered essential for the reaction. The results show that other amino acids, which possess a β-carbon (present in all amino acids except G) but lack a γ-carbon (absent in G, S, C, A), can also be used in this position.

## Discrimination between different recognition sequences

This unexpected finding allowed us to design reactions, in which educts and peptide byproducts differ in their N-terminal amino acid. This discrimination criterion could then be used to specifically inactivate the $X_1$GAFDADPLVVEI peptide byproduct (PGAFDADPLVVEI-Strep in *Figure 2*) without affecting the $X_2$GAFDADPLVVEI-**B** educt (peptide without **B** in *Figure 2*). Such selective inactivation can be achieved through methods that specifically target $X_1$ (e.g., A, C, S, P, etc.) while leaving $X_2$ ($\neq X_1$) unmodified. We hypothesized that for example an N-acetyltransferase, which acetylates the N-terminal alanine on the peptide byproduct ($X_1$=A), while leaving the N-terminal proline on educts ($X_2$=P) unmodified, might be employed to specifically inactivate the undesired byproduct (*Lapteva et al., 2021*). Another possibility is to use chemicals, which form ring structures with the amino and sulfhydryl-groups of N-terminal cysteines ($X_1$=C, $X_2$=A /P) (*Bandyopadhyay et al., 2016*). Finally, it is possible to use aminopeptidases, which act exclusively on the peptide byproduct $X_1$. Many of these enzymes have no absolute specificity for just one amino acid (*Gonzales and Robert-Baudouy, 1996*). Proline residues, however, are structurally distinct and not modified by many promiscuous enzymes, but instead by a set of proline-specific enzymes (*Cunningham and O'Connor, 1997*).

Based on these considerations, we decided to search for a proline aminopeptidase (*Dong et al., 2022*), which removes the N-terminal proline from PGAFDADPLVVEI sequences with suitable efficiency, but is inactive towards all other residues (including $X_2$=A). Literature research identified *Bacillus coagulans* proline aminopeptidase (BcPAP) as a candidate. This enzyme had been shown to cleave N-terminal proline from peptides consisting of 2–4 amino acids, while remaining inactive towards other N-terminal amino acids (*Yoshimoto and Tsuru, 1985*; *Kitazono et al., 1992*). We could produce it as a soluble monomer (33 kDa) in *E. coli* (>40 mg from 1 L culture) and tested its suitability for shifting the Connectase reaction equilibrium.

## A method for complete protein-protein fusions

We tested the effect of BcPAP in ligation reactions with **A**-ELASKDPGAFDADPLVVEI (**A**=LysS (Lysine-tRNA ligase), GST (Glutathione-S-Transferase), Ub (Ubiquitin)) and AGAFDADPLVVEI-**B** (**B**=MBP (Maltose Binding Protein), Ub, - (just the peptide)) substrates, which cover a range of molecular weights (1–61 kDa). The reactions were performed at room temperature (22 °C) in a neutral buffer (pH 7.0), with moderate salt concentrations (150 mM NaCl, 50 mM KCl), 100 μM of each substrate (A and B), as well as 0.033 eq. Connectase, and 0.066 eq. BcPAP. They were separated by SDS-PAGE, stained with Coomassie G-250, and imaged with a fluorescence scanner (Excitation 685 nm, Emission 725 nm). This allowed the densitometric quantification of the resulting protein bands with good accuracy.

In each case (*Figure 3A–C*, *Appendix 1—figure 2A*), we observed 98–100% conversion of the less abundant substrate without any reaction side products. In protein-protein ligations (*Figure 3A–B*, *Appendix 1—figure 2A*), ~90% fusion product was obtained after one hour of incubation, and ~95% after two hours. Protein-peptide ligations (*Figure 3C*) proceeded even faster. The reaction rate was approximately four times slower at low substrate concentrations (10 μM [*Figure 3D*] instead of 100 μM [*Figure 3A*]) and about eight times slower at low temperatures (10 °C [*Figure 3E*] instead of 22 °C [*Figure 3A*]), but still resulted in complete protein-protein fusions.

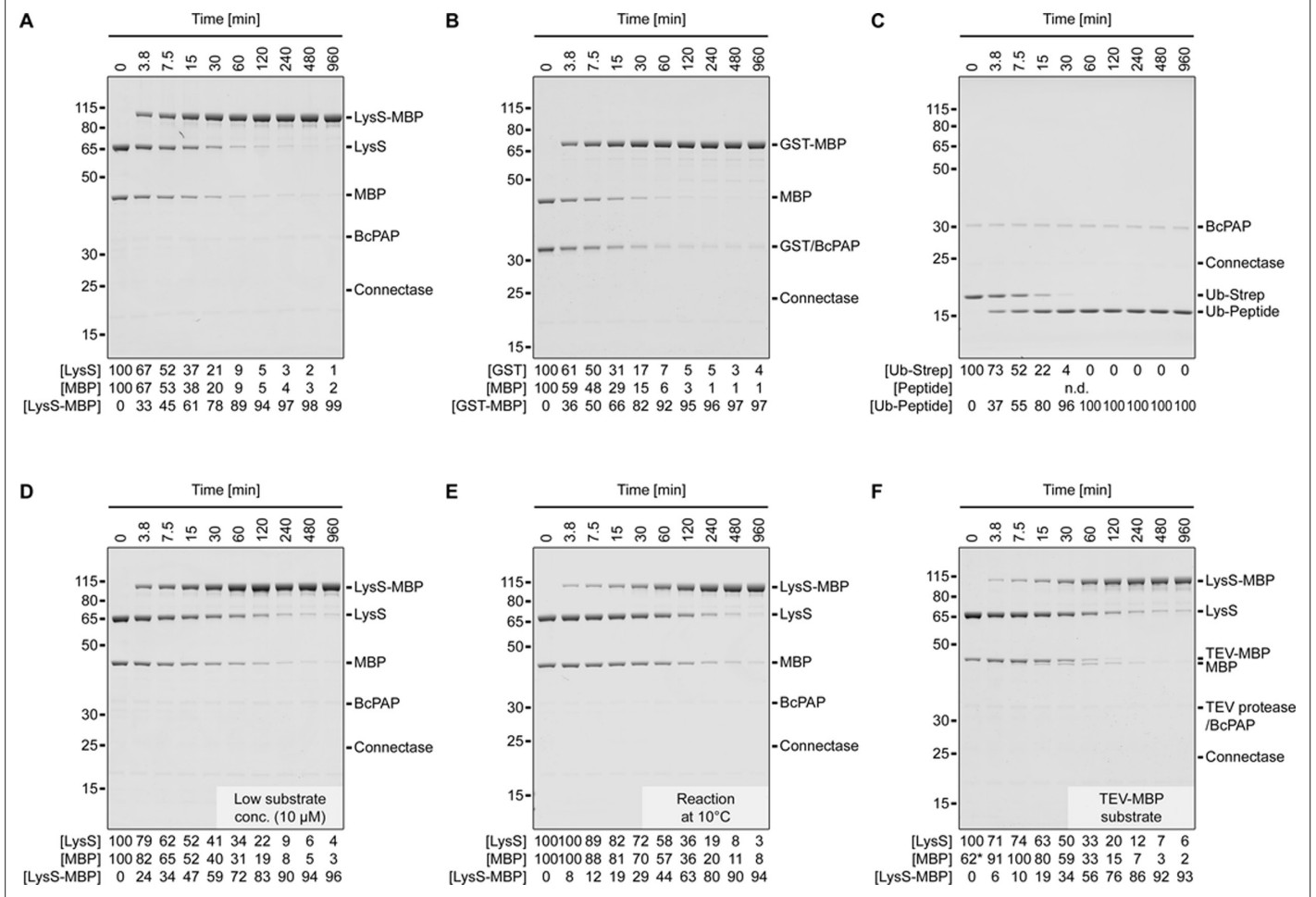

**Figure 3.** Complete protein-protein ligations. Shown are SDS-PAGE time course analyses of ligation reactions using Lysine-tRNA ligase (LysS) and Maltose binding protein (MBP; **A, D, E, F**), Glutathione-S-Transferase (GST) and MBP (**B**), or Ubiquitin with Streptavidin tag (Ub-Strep) and AGAFDADPLVVEI peptide (**C**) as substrates. Each reaction was performed with 1 eq. N-terminal fusion partner (LysS, GST or Ubiquitin; C-terminal ELASKDPGAFDADPLVVEI sequence), 1 eq. C-terminal fusion partner (MBP or peptide; N-terminal AGAFDADPLVVEI sequence), 0.033 eq. Connectase, and 0.066 eq. BcPAP. The substrate concentration was 100 µM (except for **D**: 10 µM) and the incubation temperature was 22°C (except for E: 10°C). In experiment **F**, an MBP protein with an additional N-terminal TEV protease recognition sequence (MENLYFQ|AGAFDADPLVVEI-MBP) was used and TEV protease (0.01 eq.) was added to the reaction. A densitometric analysis of the protein bands is shown below each experiment. For the substrates, the values reflect the substrate band density relative to the substrate band in the control sample (0 min); for the products, the values reflect the product band density relative to the total band density (substrates + products).

The online version of this article includes the following source data for figure 3:

**Source data 1.** PDF file containing original gel images, indicating relevant bands.

**Source data 2.** Original gel images.

**Source data 3.** Measurements and calculations.

We generated the AGAFDADPLVVEI-**B** substrates for these experiments by TEV protease cleavage of MENLYFQ|AGAFDADPLVVEI-**B** precursors. We chose this approach to avoid the potential acetylation of N-terminal alanine residues during the expression of (M)AGAFDADPLVVEI-**B** substrates (methionine removal by methionine aminopeptidase). The cleavage is efficient as the TEV protease recognition sequence is exposed N-terminally of the unstructured Connectase recognition sequence. Consequently, cleavage and conjugation can be performed in parallel with small amounts of enzyme (*Figure 3F*). Another way to prevent N-terminal acetylation is to add an extra N-terminal proline (i.e. P|AGAFDADPLVVEI-**B**), which is removed during the reaction by BcPAP (*Appendix 1—figure 2B*).

To substantiate these findings, we conducted a liquid chromatography mass spectrometry (LC-MS) analysis. The signal intensities in these experiments allow no protein quantifications, as smaller

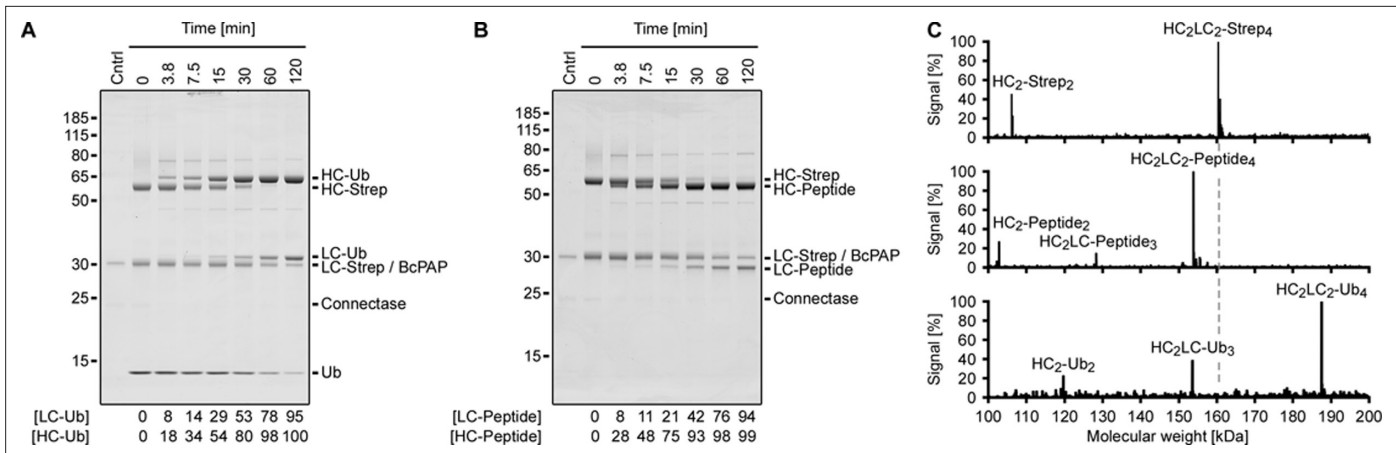

**Figure 4.** Antibody conjugation. Shown are SDS-PAGE time course (**A, B**) and LC-MS analyses (**C**) of αHER2 (human epidermal growth factor receptor 2) antibody conjugations. The αHER2 heavy (HC) and light chains (LC) were produced with a C-terminal Connectase recognition sequence and a Streptavidin tag (HC-Strep, LC-Strep). In the reactions (25 μM αHER2 (100 μM subunits), 0.033 eq. Connectase, 0.066 eq. BcPAP, 22°C), the Streptavidin tag is replaced by Ubiquitin (**A**; 1 eq.) or a shorter peptide (**B**; 1 eq.). A densitometric quantification of the product bands relative to the educt bands is shown below the gels. For the calculation, the BcPAP density in the control lane (Cntrl) Subtracted from the combined LC-Strep/BcPAP band. The LC-MS analyses (**C**) show the assemblies in the unconjugated antibody sample (top panel) and a shift of the detected masses, consistent with a near-complete conjugation to peptide (middle panel) or ubiquitin (lower panel).

The online version of this article includes the following source data for figure 4:

**Source data 1.** PDF file containing original gel images, indicating relevant bands.

**Source data 2.** Original gel images.

**Source data 3.** Measurements and calculations.

molecules are often measured with more intense signals. Nevertheless, we report relative intensities in this paper as a qualitative measure. For a 1:1 Ub-MBP conjugation (*Appendix 1—figure 3*), they amounted to 0.25% (Ub), 0.13% (MBP), and 99.6% (Ub-MBP). The N-terminal proline was almost completely (99.8%) removed from the peptide byproduct, (P)GAFDAPLVVEI. Similar to all other LC-MS tested molecules in this study (i.e. Connectase, BcPAP, MBP, LysS, αHER2 LC/HC, Ubiquitin; *Figures 4 and 5*), it was not further processed on the N-terminus. This supports the finding (*Gonzales and Robert-Baudouy, 1996*; *Yoshimoto and Tsuru, 1985*; *Kitazono et al., 1992*) that BcPAP acts exclusively on N-terminal proline residues.

Finally, we tested the effect of serine-, cysteine-, or metalloprotease inhibitors on the reaction. Connectase is not a protease (*Fuchs et al., 2021*) and therefore unaffected by these substances (*Appendix 1—figure 4*). However, the equilibrium shift associated with BcPAP activity could be suppressed with the serine protease inhibitors PMSF and AEBSF (*Appendix 1—figure 4*). These results contrast with previous studies, where BcPAP was found to be more susceptible to cysteine protease inhibitors (*Kitazono et al., 1992*). They are, however, consistent with the classification of BcPAP as a serine protease (*Kitazono et al., 1994*).

## Homogeneous antibody conjugates

Antibodies are the most relevant protein conjugation target (*Beck et al., 2017*). Many applications require their conjugation to spacious payload molecules (e.g. horseradish peroxidase) and/or to a defined number of molecules. This number, also known as drug: antibody ratio, is used as a benchmark for several specialized techniques. Of these, formyl-glycine insertion (*Liu et al., 2019*) (Catalent), sugar engineering (*Wijdeven et al., 2022*; Mersana), cysteine engineering (*Sadowsky et al., 2017*; Genentech), the introduction of unnatural amino acids (*Zimmerman et al., 2014*; Sutro), and Sortase-mediated conjugations (*Gebleux et al., 2012*; NBE) all find commercial use. Each of these approaches has its advantages and disadvantages. One problem is that even a relatively high conjugation ratio of 90% on each antibody chain leads to a desired drug: antibody ratio of 4 in only 66% of all $HC_2LC_2$ antibodies, as each antibody chain is modified separately. The completely labeled antibodies are hard to separate from the partially labeled antibodies and side reactions can further complicate the process.

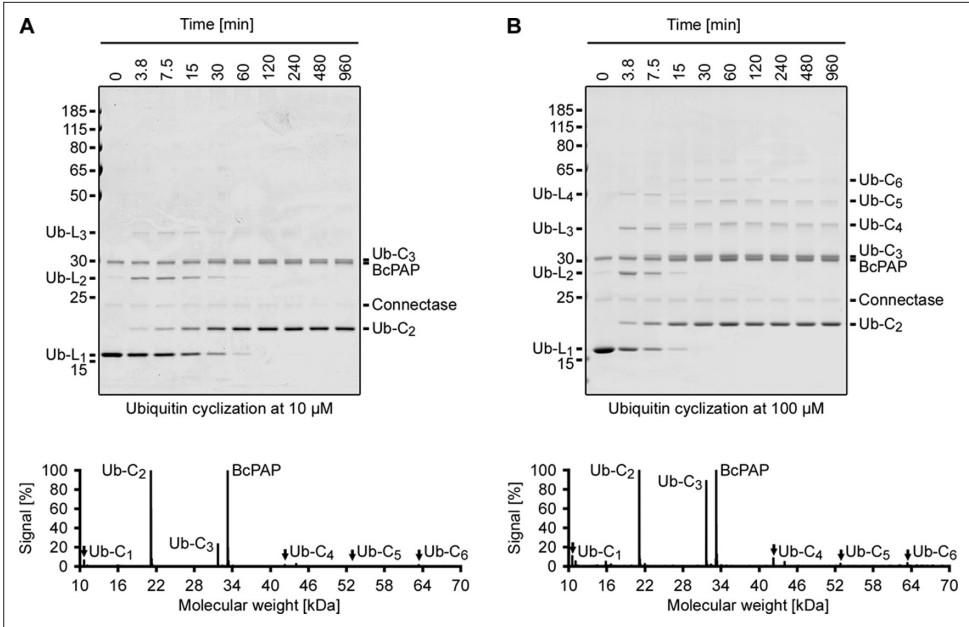

**Figure 5.** Protein cyclization. Shown are SDS-PAGE time course analyses of a Ubiquitin cyclization reaction. The employed Ubiquitin substrate was produced with both an N-terminal (AGAFADPLVVEI) and a C-terminal (ELASKDPGAFDADPLVVEI) Connectase recognition sequence. This allows the formation of linear ($L_1$ - $L_4$, formed by 1–4 Ubiquitin proteins) polymers, which are observed in the early stages of the time course. The N-terminus of a given polymer can be fused to its C-terminus, resulting in cyclic assemblies ($C_2$ - $C_6$, formed by 2–6 Ubiquitin proteins), which present the end product of the reaction. A lower substrate concentration (**A**, 10 µM) results in smaller assemblies, and a higher substrate concentration results in larger assemblies (**B**, 100 µM). The assignment of the gel bands is consistent with LC-MS data (below the gels). The plots were normalized to the most intense Ubiquitin signal (Ub-$C_2$); the BcPAP peaks are more intense (>100%), despite its relatively low abundance. The molecular masses of the ubiquitin assemblies are 13 kDa ($L_1$), 24 kDa ($L_2$), 34 kDa ($L_3$), 45 kDa ($L_4$), 21 kDa ($C_2$), 32 kDa ($C_3$), 43 kDa ($C_4$), 53 kDa ($C_5$), and 64 kDa ($C_6$).

The online version of this article includes the following source data for figure 5:

**Source data 1.** PDF file containing original gel images, indicating relevant bands.

**Source data 2.** Original gel images.

**Source data 3.** Measurements and calculations.

To test the established method for antibody conjugation, we added the Connectase recognition sequence plus an additional C-terminal Streptavidin tag to the C-termini of αHER2 light and heavy chains. HEK293 cells were transiently transfected with plasmids encoding for these proteins, and the culture medium was tested for the exported antibodies with the in-gel fluorescence method (*Fuchs, 2023*; *Fuchs, 2024*). In this western blot alternative, Connectase is used to fuse far-red fluorophores to target proteins, which can subsequently be detected on the SDS-gel with a fluorescence imager (*Appendix 1—figure 5*). From this, we expected a concentration of ~2 µg antibody per ml of culture medium – a value that could be confirmed after protein purification with a Streptavidin column.

The purified antibodies were labeled with either AGAFDADPLVVEI peptide, AGAFDADPLVVEI-Ubiquitin (1 eq. per antibody subunit; reaction conditions as in *Figure 3*), or AGAFDADPLVVEIK(-Biotin) (*Appendix 1—figure 6*). SDS-PAGE time course analyses show almost complete labeling reactions after an incubation time of two hours (*Figure 4A–B*). The heavy chain is labelled faster than the light chain, possibly because the Connectase recognition sequence is more accessible in this case. The conjugation products remained stable for several days in blood plasma (*Appendix 1—figure 7*).

The reactions were also analyzed by LC-MS. For this, the antibody carbohydrate side chains were removed enzymatically to reduce sample complexity (*Gramlich et al., 2021*). The masses determined in the unconjugated antibody sample were consistent with assemblies of completely intact antibody subunits, except for the C-terminal Strep-tag lysine, which was missing in most subunits. Of

the assigned masses, $HC_2LC_2$ assemblies constituted 69% of the signal, followed by $HC_2$ (30%), and $HC_2LC_1$ (0.4%) assemblies (*Figure 4C*, top panel).

Next, a 1:1 antibody-peptide conjugation reaction was analyzed with the same method (*Figure 4C*, middle panel). Here, the $HC_2LC_1$ assembly was detected only in completely conjugated form, as $HC_2LC_1$-peptide$_3$ (100%). Similarly, $HC_2$ was completely conjugated to $HC_2$-peptide$_2$ (100%). $HC_2LC_2$ was found to be conjugated to either two (2%), three (15%), or four (83%) peptides. Taken together, this suggests a total labeling efficiency of 97%, in line with the results shown in *Figure 4B*.

Finally, a 1:1 antibody-ubiquitin conjugation was analyzed (*Figure 4C*, lower panel). Here, the $HC_2LC_1$ assembly was more abundant and the $HC_2$ assembly less abundant compared to the unconjugated sample. Again, $HC_2$ and $HC_2LC_1$ were detected only in completely conjugated form, as $HC_2$-$Ub_2$ (100%) and $HC_2LC_1$-$Ub_3$ (100%). $HC_2LC_2$ was found to be conjugated to either one (2%), two (1%), three (2%), or four (95%) ubiquitin molecules, suggesting a total labeling efficiency of 98%.

These results demonstrate that the method is effective for the rapid generation of near-homogeneous antibody conjugates. The conjugation ratio could potentially be further increased with a small excess of the conjugation partner (here: Ubiquitin or peptide). The desired products can be purified in subsequent steps, for example with Streptavidin columns to remove unconjugated antibodies, or with size exclusion columns, to separate $HC_2LC_2$ from other assemblies.

## Protein cyclization and polymerization

Protein cyclization is used to enhance protein stability (*Purkayastha and Kang, 2019*), while protein polymerization can be useful for engineering biomaterials or affine binders (*Fierer et al., 2014*). Both results may be achieved with proteins carrying both N- and C-terminal Connectase recognition sequences. Circularization may be favored, when the protein is present at a low concentration and when its N- and C-termini are in close proximity. Polymerization might be favored for rod-shaped proteins with distant N- and C-termini at high concentrations.

To test these ideas, we used Ubiquitin (AGAFDADPLVVEI-Ub-ELASKDPGAFDADPLVVEI) as a small globular test substrate at low concentrations (10 µM, *Figure 5A*). At the start of the reaction (0–7.5 min), we observed its polymerization to linear chains of two (Ub-$L_2$) or three (Ub-$L_3$) protomers. These linear assemblies gradually disappeared (7.5–120 min), as their N- and C-termini were fused. The resulting circular assemblies (80% Ub-$C_2$ and 20% Ub-$C_3$, according to SDS-PAGE densitometry) presented the stable end-products of the reaction.

When the reaction was performed with ten-fold higher substrate concentrations (100 µM, *Figure 5B*), longer linear ubiquitin polymers were initially formed, which eventually gave rise to larger circular assemblies (37% Ub-$C_2$, 45% Ub-$C_3$, 9% Ub-$C_4$, 6% Ub-$C_5$, and 2% Ub-$C_6$). Single ubiquitin molecules were not efficiently circularized in either experiment, suggesting that the N- and C-termini in one ubiquitin molecule are too far apart to be connected, and that additional linker sequences are needed to favor this assembly.

Taken together, the results show that the presented conjugation method enables efficient protein circularizations and that the yields of specific assemblies can be controlled by modifying the reaction conditions (e.g. substrate concentration) and the substrate design (e.g. linker length). As with other cyclization/polymerization methods (*van 't Hof et al., 2015*), these parameters need to be tested for each individual substrate.

## Discussion

In this paper, we have shown that the Connectase recognition sequence can be altered, so that the peptide reaction byproduct can be specifically inactivated by a proline aminopeptidase. This method enables the fast, simple, specific, and complete 1:1 fusion of proteins and/or peptides.

Instead of just fusing two molecules, as for example in chemical ligations, the method replaces one sequence (e.g. PGAFDADPLVVEI-Strep in *Figure 4*) with the fusion partner. This has several advantages. For example, an affinity tag used for substrate purification can be removed by the conjugation reaction. As the affinity tag remains on unconjugated substrates, it can subsequently be used to remove the peptide byproduct and any unconjugated substrates (e.g. an antibody conjugated to only three molecules instead of four). When the same affinity tag is added to Connectase and BcPAP, this setup enables the complete purification of homogeneous fusion product in a single step (*Figure 6*).

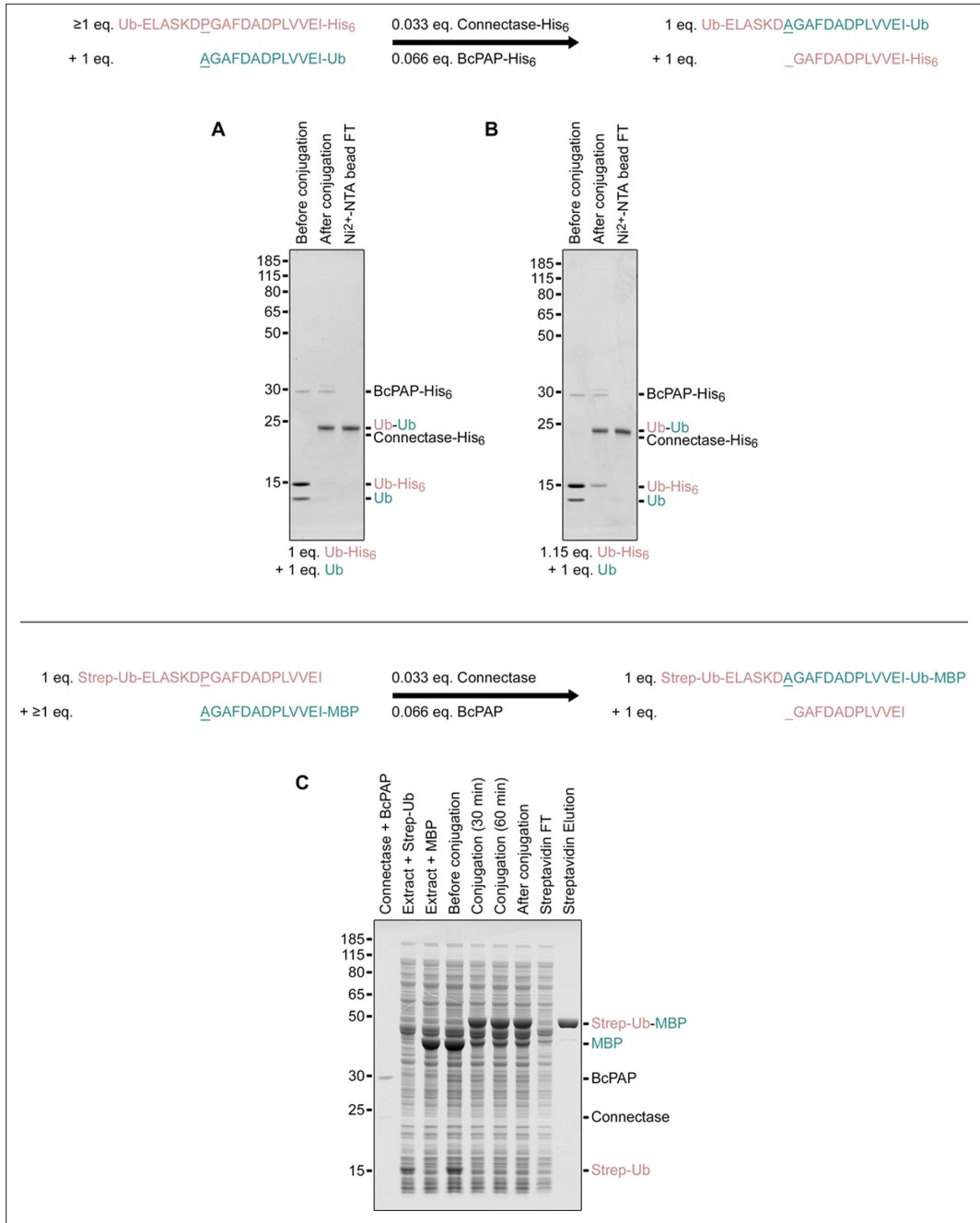

**Figure 6.** Two exemplary strategies for purifying protein conjugates. (**A, B**) The first strategy uses purified educts and couples the conjugation reaction with affinity tag removal (top scheme). This allows the separation of the untagged product (Ub-Ub conjugate) from tagged enzymes (Connetase-His$_6$, BcPAP-His$_6$), side products (GAFDADPLVVEI-His$_6$ peptide), and the first educt (Ub-His$_6$) by collecting the unbound fraction (flow through (FT)) in an affinity chromatography step. The second untagged educt (AGADADPLVVEI-Ub) must be entirely converted. This can be achieved with sufficient incubation times in equimolar reactions (**A**) or by using a small excess of reaction partner (**B**). (**C**) The second strategy can be used for expressed proteins in cell extracts and introduces an affinity tag in the reaction product (top scheme). It necessitates the complete conversion of the first affinity tagged educt (Strep-Ub). This is most conveniently achieved by using the second untagged educt (MBP) in excess. Note that proteins with N-terminal alanine residues, such as AGAFDADPLVVEI-MBP, may be partially N-acetylated in some expression systems. In these cases, a higher excess (e.g. more extract with MBP) must be used, as acetylated substrates are not converted by Connectase. Strategies to avoid N-acetylation are described in *Figure 3* and

*Figure 6 continued on next page*

*Figure 6 continued*

**Appendix 1—figure 2**. Aside from these two strategies, alternative methods such as tandem affinity purification or size exclusion chromatography can also be employed.

The online version of this article includes the following source data for figure 6:

**Source data 1.** PDF file containing original gel images, indicating relevant bands.

**Source data 2.** Original gel images.

Another consequence of this replacement mechanism is that complete fusions may be reversed on demand. This is not possible with chemical, split intein, or split domain ligation methods. Yet, the addition of a new $X_3$GAFDADPLVVEI-**C** fusion partner to an existing **A**-ELASKDX$_2$GAFDADPLVVEI-**B** fusion product allows the exchange of molecules **B** and **C**. To make this second conjugation complete, a new method for the specific removal of $X_2$ is needed, just like BcPAP was used to remove $X_1$=Pro in this study. This new method could involve another aminopeptidase (*Gonzales and Robert-Baudouy, 1996*), a specific chemical modification (*Bandyopadhyay et al., 2016*), or an N-acetyltransferase (*Lapteva et al., 2021*) (see the section 'Discrimination between different recognition sequences'). The resulting re-conjugation would enable advanced applications, such as the immobilization of a protein on a surface and its release on demand, or the visualization of existing conjugation products by fluorophore (de-) coupling.

These considerations illustrate the versatility of Connectase as a bioengineering tool. As more applications with this enzyme are developed, synergies will emerge. For example, in this paper, the same Connectase recognition sequence could be used to monitor antibody expression levels in a western blot-like application (*Appendix 1—figure 5*) and for the subsequent antibody conjugation. In future, other Connectase applications, such as protein purification or microplate-based protein quantification, may offer additional functionality for the recognition sequence.

During the preparation of this study, a somewhat related method was published for the enzyme Sortase A (*Arnott et al., 2024*). The authors fused various sequences, such as **A**-LPETGAHHHHH and GVGSKYG for **A**-LPETGVGSKYG generation (C-terminal labeling) or YALPETGG and GVGK-**B** for YALPETGVGK-**B** generation (N-terminal labeling). The peptide byproducts (i.e. GAHHHHHH or GG), were digested with an aminopeptidase. This aminopeptidase showed a preference for GG or GA over GV and therefore displayed a lower activity towards GVGSKYG or GVGK-**B** substrates. The method significantly increased the obtained fusion product yields and therefore presents an important development for Sortase-mediated ligations. Despite these similarities, there are important differences compared to the method presented here. As described in the introduction, Sortase irreversibly hydrolyzes both educts and products, shows low specificity for the C-terminal fusion partner, and has low catalytic efficiency (*Schmidt et al., 2017*; *Morgan et al., 2022*). As a result, the method is most efficient for C-terminal protein conjugations with small peptides at high concentrations (*Arnott et al., 2024*). Protein-protein conjugations are relatively inefficient, as they result in high quantities of hydrolysis side products and remain largely incomplete, even after several days of incubation with high enzyme quantities.

The main disadvantage of Connectase lies in its relatively long recognition sequence. The substrates require a 13 amino acid recognition sequence on the N-terminus and a 19 amino acid sequence on the C-terminus (*Appendix 1—figure 8*). For protein-protein conjugations, we employed additional linker sequences of one (N-terminal R) and five (C-terminal AAAGA) amino acids. This design resulted in efficient conjugations for each substrate tested so far, including 44 constructs derived from 19 different proteins (*Fuchs et al., 2021*; *Fuchs, 2023*; *Fuchs, 2024*). Therefore, the employed linkers can generally be regarded as sufficient, even in cases of poor steric accessibility of the substrate termini. Taken together, this results in a final conjugation 'scar' of 19–25 amino acids between the fusion partners. This is comparable to the SpyTag / KTag system (23 amino acids plus linker sequences; *Fierer et al., 2014*), but longer than the sequences typically employed for Sortase-mediated fusions (7–9 amino acids, usually with additional (GGGGS)$_x$ linkers between the protein(s) and the ligation site *Antos et al., 2017*). Long linker sequences can potentially interfere with protein folding, and they can be susceptible to proteases, or proof immunogenic. Although these problems have not been encountered for Connectase so far, it is an attractive option to engineer Connectase variants with a shorter recognition sequence in the future. For now, the method described here is primarily useful in

applications, where this longer 'scar' can be tolerated. In these cases, however, the combination of simplicity, substrate specificity, efficiency, and completeness, along with the possibility to use either biologically or chemically produced substrates, and the absence of side reactions, is unmatched by other chemical or enzymatic methods.

# Methods

## Key resources table

| Reagent type (species) or resource | Designation | Source or reference | Identifiers | Additional information |
|---|---|---|---|---|
| Strain, strain background (*Escherichia coli*) | BL21(DE3) gold | Agilent | #230130 | - |
| Strain, strain background (*Homo sapiens*) | HEK293 | Sigma-Aldrich | 85120602 | - |
| Recombinant DNA reagent | Connectase-GS-His$_6$ | synthesized by Biocat (this paper) | - | *Supplementary file 1* |
| Recombinant DNA reagent | BcPAP-GS-His$_6$ | synthesized by Biocat (this paper) | - | *Supplementary file 1* |
| Recombinant DNA reagent | His$_6$-GS-Ubiquitin-RELASKD-CnTag-SEEGEGS-Strep | synthesized by Biocat (this paper) | - | *Supplementary file 1* |
| Recombinant DNA reagent | His$_6$-GS-LysS-RELASKD-CnTag | synthesized by Biocat (this paper) | - | *Supplementary file 1* |
| Recombinant DNA reagent | His$_6$-GS-GST-RELASKD-CnTag | synthesized by Biocat (this paper) | - | *Supplementary file 1* |
| Recombinant DNA reagent | His$_6$-TEV-CnTag(P1A)-AAAGA-MBP | synthesized by Biocat (this paper) | - | *Supplementary file 1* |
| Recombinant DNA reagent | His$_6$-GS-TEV protease | synthesized by Biocat (this paper) | - | *Supplementary file 1* |
| Recombinant DNA reagent | His$_6$-TEV-CnTag(P1A)-AAAGA-Ubiquitin | synthesized by Biocat (this paper) | - | *Supplementary file 1* |
| Recombinant DNA reagent | Pro-CnTag(P1A)-AAAGA-Ubiquitin-His$_6$ | synthesized by Biocat (this paper) | - | *Supplementary file 1* |
| Recombinant DNA reagent | αHER2-Heavy chain-RELASKD-CnTag-SEEGEGSG-Strep | synthesized by Biocat (this paper) | - | *Supplementary file 1* |
| Recombinant DNA reagent | αHER2-Light chain-RELASKD-CnTag-SEEGEGSG-Strep | synthesized by Biocat (this paper) | - | *Supplementary file 1* |
| Recombinant DNA reagent | His$_6$-TEV-CnTag(P1A)-AAAGA-Ubiquitin-RELASKD-CnTag-SE-Strep | synthesized by Biocat (this paper) | - | *Supplementary file 1* |
| Recombinant DNA reagent | Ub-RELASKD-CnTag-His$_6$ | synthesized by Biocat (this paper) | - | *Supplementary file 1* |
| Recombinant DNA reagent | Strep-Ub-RELASKD-CnTag | synthesized by Biocat (this paper) | - | *Supplementary file 1* |
| Peptide, recombinant protein | AGAFDADPLVVEI | synhesized by Intavis (this paper) | - | *Supplementary file 1* |
| Peptide, recombinant protein | CGAFDADPLVVEI | synhesized by Intavis (this paper) | - | *Supplementary file 1* |
| Peptide, recombinant protein | DGAFDADPLVVEI | synhesized by Intavis (this paper) | - | *Supplementary file 1* |
| Peptide, recombinant protein | EGAFDADPLVVEI | synhesized by Intavis (this paper) | - | *Supplementary file 1* |
| Peptide, recombinant protein | FGAFDADPLVVEI | synhesized by Intavis (this paper) | - | *Supplementary file 1* |
| Peptide, recombinant protein | GGAFDADPLVVEI | synhesized by Intavis (this paper) | - | *Supplementary file 1* |

*Continued on next page*

*Continued*

| Reagent type (species) or resource | Designation | Source or reference | Identifiers | Additional information |
|---|---|---|---|---|
| Peptide, recombinant protein | HGAFDADPLVVEI | synhesized by Intavis (this paper) | - | *Supplementary file 1* |
| Peptide, recombinant protein | IGAFDADPLVVEI | synhesized by Intavis (this paper) | - | *Supplementary file 1* |
| Peptide, recombinant protein | KGAFDADPLVVEI | synhesized by Intavis (this paper) | - | *Supplementary file 1* |
| Peptide, recombinant protein | LGAFDADPLVVEI | synhesized by Intavis (this paper) | - | *Supplementary file 1* |
| Peptide, recombinant protein | MGAFDADPLVVEI | synhesized by Intavis (this paper) | - | *Supplementary file 1* |
| Peptide, recombinant protein | NGAFDADPLVVEI | synhesized by Intavis (this paper) | - | *Supplementary file 1* |
| Peptide, recombinant protein | PGAFDADPLVVEI | synhesized by Intavis (this paper) | - | *Supplementary file 1* |
| Peptide, recombinant protein | QGAFDADPLVVEI | synhesized by Intavis (this paper) | - | *Supplementary file 1* |
| Peptide, recombinant protein | RGAFDADPLVVEI | synhesized by Intavis (this paper) | - | *Supplementary file 1* |
| Peptide, recombinant protein | SGAFDADPLVVEI | synhesized by Intavis (this paper) | - | *Supplementary file 1* |
| Peptide, recombinant protein | TGAFDADPLVVEI | synhesized by Intavis (this paper) | - | *Supplementary file 1* |
| Peptide, recombinant protein | VGAFDADPLVVEI | synhesized by Intavis (this paper) | - | *Supplementary file 1* |
| Peptide, recombinant protein | WGAFDADPLVVEI | synhesized by Intavis (this paper) | - | *Supplementary file 1* |
| Peptide, recombinant protein | YGAFDADPLVVEI | synhesized by Intavis (this paper) | - | *Supplementary file 1* |
| Peptide, recombinant protein | Peptide for in-gel fluorescence detection | synhesized by Intavis (this paper) | - | *Supplementary file 1* |
| Peptide, recombinant protein | Peptide for biotinylation | synhesized by Intavis (this paper) | - | *Supplementary file 1* |
| Peptide, recombinant protein | RELASKDPGAFDADPLVVEI | synhesized by Intavis (this paper) | - | *Supplementary file 1* |
| Peptide, recombinant protein | ELASKDPGAFDADPLVVEI | synhesized by Intavis (this paper) | - | *Supplementary file 1* |
| Peptide, recombinant protein | LASKDPGAFDADPLVVEI | synhesized by Intavis (this paper) | - | *Supplementary file 1* |
| Commercial assay or kit | Protease Inhibitor kit | G-biosciences | 786–207 | - |
| Chemical compound, drug | Lipofectamine 2000 | Thermo Fisher | 11668027 | |
| Chemical compound, drug | Protease Inhibitor Cocktail | Roche | 11697498001 | |
| Software, algorithm | Unicorn v5.1.0 | GE | - | - |
| Software, algorithm | Image Studio Lite 5.2. | Licor | RRID:SCR_013715 | - |
| Software, algorithm | Compass DataAnalysis v6.1 software | Bruker | - | - |
| Software, algorithm | MaxEnt | Bruker | - | - |

## Cloning, expression, and purification

The sequences of all proteins and peptides used in this study are listed in *Supplementary file 1*. The peptide for the in-gel fluorescence assay was synthesized by Intavis, while all other peptides were synthesized by Genecust. Genes were synthesized by Biocat, and cloned into the pET30b(+) vector

(restriction sites: NdeI, XhoI) for expression in *E. coli* or the pcDNA3.1 vector (restriction sites: HindIII, XhoI) for expression in HEK293 cells.

For recombinant expression in *E. coli*, BL21 gold cells were transfected with the respective plasmids and grown in lysogeny broth medium with 50 µg/l kanamycin at 22 °C. Protein expression was induced at an optical density of 0.4 at 600 nm with 500 µM isopropyl-β-D-thiogalactoside. Cells expressing soluble proteins were harvested after 16 hr, resuspended in buffer (100 mM Tris-HCl, 1 x c0mplete EDTA-free protease inhibitor cocktail (Roche; cat. # 11697498001; no inhibitor was added to cells expressing BcPAP), 0.02 g/l DNAse, pH 8.0), lysed by French press, and cleared from cell debris by ultracentrifugation (120000 × *g*, 45 min, 4 °C).

For recombinant expression of αHER2 antibodies, HEK293 cells were cultured at 37 °C in ten 75 cm² flasks with Dulbecco's Modified Eagle Medium (DMEM; cat. # 11966025) supplemented with fetal calf serum. At 70% confluency, they were transfected with plasmids encoding for αHER2 light and heavy chains using Lipofectamine 2000 (Thermo; cat. # 11668027), according to the manufacturer's instructions (47 µl Lipofectamine, 55 µg of each plasmid). The cells were grown for ten days without splitting, and the medium was exchanged daily. The medium samples were used for αHER2 detection by in-gel fluorescence (*Appendix 1—figure 5*, described below), then pooled, centrifuged (6000 × *g*, 10 min, 4 °C), and filtered (0.45 µm) before protein purification.

For protein purification, His$_6$-tagged proteins (all proteins except for antibody subunits *Figure 4*, *Appendix 1—figure 5* and *Appendix 1—figure 7*) and Strep-tagged Ubiquitin constructs (*Figure 5* and *Figure 6*) were applied to HisTrap HP columns (20 mM Tris-HCl pH 8.0, 250 mM NaCl, 20–250 mM imidazole). Strep-tagged proteins (the antibody subunits and the Ubiquitin constructs) were instead purified with StrepTrap XT columns (1.8 mM KH$_2$PO$_4$, 10 mM Na$_2$HPO$_4$, 2.7 mM KCl, 138 mM NaCl, 0–50 mM Biotin, pH 7.4). After this initial purification step, proteins with N-terminal TEV recognition sequences (MBP (*Figure 3* and *Appendix 1—figure 3*)), Ubiquitin (*Figure 4* and *Appendix 1—figure 2*), and Ubiquitin for cyclization (*Figure 5*) were incubated with TEV protease at a 1:100 molar ratio. The reaction was performed overnight in dialysis tubes (dialysis buffer: 20 mM Tris-HCl pH 8.0, 250 mM NaCl) at 4 °C. The processed proteins were separated from His6-tagged TEV protease, N-terminal fragments (MHHHHHHENLYFQ), and residual unprocessed proteins by another purification step. For this, the reactions were applied a second time to HisTrap HP columns (as above), and the flow-through was collected. All chromatography steps were performed on an Äkta Purifier FPLC (GE Healthcare) using Unicorn v5.1.0 software. Purified proteins were supplemented with 15% glycerol, flash-frozen in liquid nitrogen, and stored at –80 °C.

## Biochemical ssays

Unless noted otherwise, all conjugation reactions were performed at 22 °C in neutral (pH 7.0) buffer containing 50 mM sodium acetate, 50 mM MES, 50 mM HEPES, 150 mM NaCl, and 50 mM KCl. They were stopped with SDS loading buffer (final concentration: 50 mM Tris-HCl, 2% SDS, 10% glycerol, 25 mM β-mercaptoethanol, 0.01% bromophenol blue, pH 6.8; final protein concentration ~0.1 g/l) and incubated at 90 °C for 10 min. The samples were separated using mPAGE 12% Bis-Tris gels (Merck; 5 µl loading volume per sample) with MOPS running buffer (50 mM MOPS, 50 mM Tris, 0.1% SDS, 1 mM EDTA; no pH adjustment). The gels were stained with Coomassie blue (25% ethanol, 25% methanol, 10% acetate, 0.25% Coomassie R-250), and subsequently with Coomassie colloidal solution (20% ethanol, 10% ammonium sulfate, 5.8% phosphoric acid, 5% methanol, 0.12% Coomassie G-250). They were destained with 10% acetic acid and imaged with an Azure Sapphire NIR fluorescence scanner (excitation at 685 nm, emission at 725 nm, 25–50 µm resolution, Intensity 8, highest scanning speed). Densitometric band quantification was performed with Image Studio Lite 5.2. The bands were manually assigned with the 'draw rectangle' tool. Each quantified region encompassed the entire band, bordered by a minimum of three pixels of background for accurate measurement.

For *Figure 2*, 20 experiments were set up, each with a different XGAFDADPLVVEI peptide (X=any of the 20 amino acids). The reactions contained 20 mM Ub-Strep, 20 mM peptide, and 0.2 mM Connectase. Samples were taken before the addition of Connectase (0 min) and after the indicated times (0.1–96 hr). After SDS-PAGE and densitometric analyses, the relative reaction rates in each experiment were estimated. An exact determination was not possible because the reaction is reversible, and the emerging peptide side product (PGAFDADPLVVEI-Strep) competes with the assayed

peptide (XGAFDADPLVVEI) for the enzyme binding sites. The estimates were made based on the required time to obtain 10%, 20%, and 30% fusion product yield in each reaction.

For *Figure 3* and *Appendix 1—figure 2*, 8 experiments were set up with different substrate pairs (*Figure 3A, D and E*: LysS/MBP; *Figure 3B*: GST/MBP; *Figure 3C*: Ub-Strep/AGAFDADPLVVEI peptide; *Figure 3F*: LysS/MBP before TEV cleavage (see "Purification"); *Appendix 1—figure 2A*: LysS/Ub; *Figure 2B*: LysS/Pro-Ub; protein sequences are listed in *Supplementary file 1*). Each substrate was used at 100 μM (except for *Figure 3D* (10 μM)). The reactions were started by addition of 0.033 molar equivalents (eq.) Connectase and 0.066 eq. BcPAP. For the experiment shown in *Figure 3F*, 0.01 eq. TEV protease was also added. The reactions were incubated at 22 °C (except for *Figure 3E* [10 °C]). Samples were taken before (0 min) and after the addition of Connectase/BcPAP (3.8–960 min). After SDS-PAGE, the substrate band densities relative to the control sample (0 min) were determined. The product band densities were determined relative to the total band density (substrates +products).

For *Appendix 1—figure 4*, Connectase (2 μM) was used without (A) or with BcPAP (7 μM, B). Both solutions were incubated at 22 °C for 30 min with the following compounds: buffer (control reaction), 1 mM ZnCl2, 'Complete' protease inhibitor mix (Roche, 1 tablet per 25 ml), AEBSF, ALLN, Antipain, Aprotinin, Bestatin, Chymostatin, E-64, EDTA, Leupeptin, Pepstatin, Phosphoramidon, PMSF. The concentrations of the last compounds are unknown to us, as the supplier (G-biosciences, Protease Inhibitor Set, cat. # 786–207; inhibitors used at '2 x' concentration) refused to provide them on request. After the incubation, the enzyme-inhibitor mixture was added to an equal volume of conjugation substrates (20 μM Ub-Strep, 20 μM AGAFDADPLVVEI peptide). After 2 hr at 22 °C, the reactions were analyzed by SDS-PAGE.

For *Appendix 1—figure 5*, the cell culture medium samples taken during αHER2 expression (see above) were analyzed. They were centrifuged (1 min, 10,000 × *g*) and 2.5 μl of each supernatant was mixed with 300 fmol reference protein (MBP with a C-terminal Connectase recognition sequence). The mixture was incubated with 1 nM Connectase and 10 nM fluorescent peptide substrate (RELA SKDPGAFDADPLVVEISEEGE-Cy5.5) for 20 min at 22 °C. The reactions were separated by SDS-PAGE and imaged before and after Coomassie staining. The band densities corresponding to reference protein and αHER light (LC) and heavy chains (HC) were determined. They were used to estimate the expressed αHER2 quantities over time with [αHER2]=reference protein quantities x ([signal HC] + [signal LC] / [signal reference protein]). This estimation can be turned into an exact determination of αHER2 quantities by division with an experimentally determined factor, which describes the relative reactivity and brightness of antibody and reference protein bands. In this case, this factor was close to 1, so that estimated and determined values were nearly identical. The determination and the detailed method are described in *Fuchs, 2024*.

For *Figure 4* and *Appendix 1—figure 6*, 25 μM Strep-tagged αHER2 antibody was mixed with 3.33 μM Connectase, 6.66 μM BcPAP, and either 100 μM AGAFDADPLVVEI-Ubiquitin, 100 μM AGAF-DADPLVVEI peptide (*Figure 4*), or 100 μM AGAFDADPLVVEIK(-Biotin) (*Appendix 1—figure 6*). The reactions were incubated at 22 °C, and samples were taken after the indicated times (0–120 min). After SDS-PAGE, the product band quantities (i.e., LC-Ub, HC-Ub, or LC-peptide, HC-peptide) were determined relative to the educt band quantities (LC-Strep, HC-Strep).

For *Appendix 1—figure 7*, αHER2 antibodies were biotinylated as described above and mixed (1 μM) with human plasma (a gift from Dr. Konnerth; supplemented with EDTA to prevent clotting). The mixture was divided into three tubes and incubated at 4 °C, 22 °C, or 37 °C. Samples were taken after 0, 1, 3, 5, and 7 days and stored at –20 °C. For the analysis, 2 μl of each sample was mixed with 33 μl labeling solution (20 nM Connectase, 2 nM PGAFDADPLVVEISEEGE-Cy5.5 peptide) and incubated for 30 min. The samples were then supplemented with 15 μl SDS loading buffer and separated by SDS-PAGE (5 μl per lane). The gel was first analyzed by in-gel fluorescence (see above *Appendix 1—figure 5*) and subsequently stained with Coomassie G-250.

For *Figure 5*, a ubiquitin variant with an N-terminal (AGAFDADPLVVEI...) and a C-terminal (ELAS-KDPGAFDADPLVVEI) Connectase recognition sequence was employed. This substrate was used at a concentration of 10 μM (first experiment) and at a concentration of 100 μM (second experiment). Both reactions were conducted with 0.033 eq. Connectase and 0.066 eq. BcPAP at 22 °C. Samples were taken after the indicated times and analyzed by SDS-PAGE.

For *Figure 6*, 100 μM (A) or 115 μM (B) Ubiquitin with C-terminal Connectase recognition sequence was mixed with 100 μM Ubiquitin with N-terminal Connectase recognition sequence,

6.66 µM BcPAP, and 3.33 µM Connectase. The mixtures were incubated overnight (A) or for 2 hr (B), and applied to a Ni²⁺-NTA column (see purification). Samples were collected at the start of the reaction, after the incubation, and after the chromatography step (flow-through), and analyzed by SDS-PAGE. For *Figure 6C*, Streptavidin-tagged Ubiquitin with N-terminal Connectase recognition sequence and processed MBP (N-terminal AGAFDADPLVVEI sequence, see *Figure 3*) were employed. Both proteins were mixed with *E. coli* cell extract (lane 2, 3), combined (final concentrations: ~1 g/l Ubiquitin (~80 µM), ~4 g/l MBP (~100 µM), ~40 g/l *E. coli* protein), incubated for 90 min at 22 °C with 6.66 µM BcPAP and 3.33 µM Connectase (lanes 4–6), and applied to a Streptavidin column (see purification). The described samples and the eluted conjugate were analyzed by SDS-PAGE.

For *Appendix 1—figure 8*, Ub-Strep (10 µM) was mixed with RELASKDPGAFDADPLVVEI, ELASKDPGAFDADPLVVEI, or LASKDPGAFDADPLVVEI peptides (10 µM) and 0.25 µM Connectase. Reaction samples were taken after the indicated times (0–60 min) and analyzed by SDS-PAGE.

## Liquid chromatography-mass spectrometry (LC-MS)

LC-MS analysis was performed at the Natural and Medical Sciences Institute (NMI, Reutlingen, Germany), using established sample preparation and data interpretation protocols (*Gramlich et al., 2021*; *Gramlich et al., 2022*). Specifically, the samples were prepared as follows:

For *Appendix 1—figure 3*, 100 µM Ub-Strep, 100 µM MBP, 3.33 µM Connectase, and 6.66 µM BcPAP were mixed. The reaction was incubated for 4 hr at 22 °C and then used for LC-MS analysis.

For *Figure 4*, 25 µM Strep-tagged αHER2 antibody was mixed with 3.33 µM Connectase, 6.66 µM BcPAP, and either 100 µM AGAFDADPLVVEI-Ubiquitin or 100 µM AGAFDADPLVVEI peptide. The reaction was incubated for 4 hr at 22 °C and then used for LC-MS analysis. Unconjugated αHER2 antibody was used as a control. All samples were deglycosylated with PNGase F (R&D Systems) for 16 hr at 37 °C.

For *Figure 5*, a ubiquitin variant with N- and C-terminal Connectase recognition sequence was used at two different concentrations, 10 µM and 100 µM. Both samples were incubated with 0.033 eq. Connectase and 0.066 eq. BcPAP for 4 hr at 22 °C and then used for LC-MS analysis.

The samples stored at 4 °C for up to 6 hr, before they were subjected (1.6 µg protein) to an Acquity BEH C4 column, using an UltiMate3000 UHPLC. They were eluted with a 0–50% $H_2O$ / acetonitrile gradient in presence of 0.1% formic acid over 7 min. The eluted molecules were analyzed with a MaXis HD UHR q-TOF spectrometer. Mass spectrometer parameters were adapted to the size of the molecule and the chromatography flow rate (by default 0.15 ml/min). Data analysis was performed using Bruker Compass DataAnalysis v6.1 software (Bruker Daltonik, Bremen, Germany). Charge deconvolution of the m/z spectra was performed with the MaxEnt deconvolution algorithm (Bruker Daltonic, Bremen, Germany). Deconvolution artifacts without m/z series were excluded.

## Acknowledgements

Liquid chromatography-mass spectrometry was performed by Sandra Maier and Dr. Anne Zeck at the NMI Natural and Medical Sciences Institute at the University of Tübingen, using instrumentation acquired through the Baden-Württemberg Ministry of Economic Affairs, Labor and Tourism (Germany) grant program " Special investment program for climate-neutral business-related research" (WM3-4332-3/6). We thank Andrei Lupas for discussions and continuous support. We thank Valeria Hatskovska for support in handling the eukaryotic cell cultures. This work was supported by institutional funds from the Max Planck Society and by the German Research Foundation (DFG project number 512378754 to ACDF).

## Additional information

### Competing interests

Adrian CD Fuchs: Max Planck Innovation has filed a provisional patent on the method described in this paper (EP-Patent Application EP241884741) The author declares no other competing interests.

### Funding

| Funder | Grant reference number | Author |
|---|---|---|
| Deutsche Forschungsgemeinschaft | 512378754 | Adrian CD Fuchs |

The funders had no role in study design, data collection and interpretation, or the decision to submit the work for publication. Open access funding provided by Max Planck Society.

### Author contributions

Adrian CD Fuchs, Conceptualization, Resources, Data curation, Formal analysis, Funding acquisition, Validation, Investigation, Visualization, Methodology, Writing – original draft, Project administration, Writing – review and editing

### Author ORCIDs

Adrian CD Fuchs ⬤ https://orcid.org/0000-0001-6550-1795

Reviewer #1 (Public review): https://doi.org/10.7554/eLife.102765.3.sa1
Reviewer #2 (Public review): https://doi.org/10.7554/eLife.102765.3.sa2
Author response https://doi.org/10.7554/eLife.102765.3.sa3

## Additional files

### Supplementary files

MDAR checklist

Supplementary file 1. Proteins and peptides used in the study.

### Data availability

All primary data are provided as source data files in the article. All relevant materials are available from the corresponding author upon request.

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

# Appendix 1

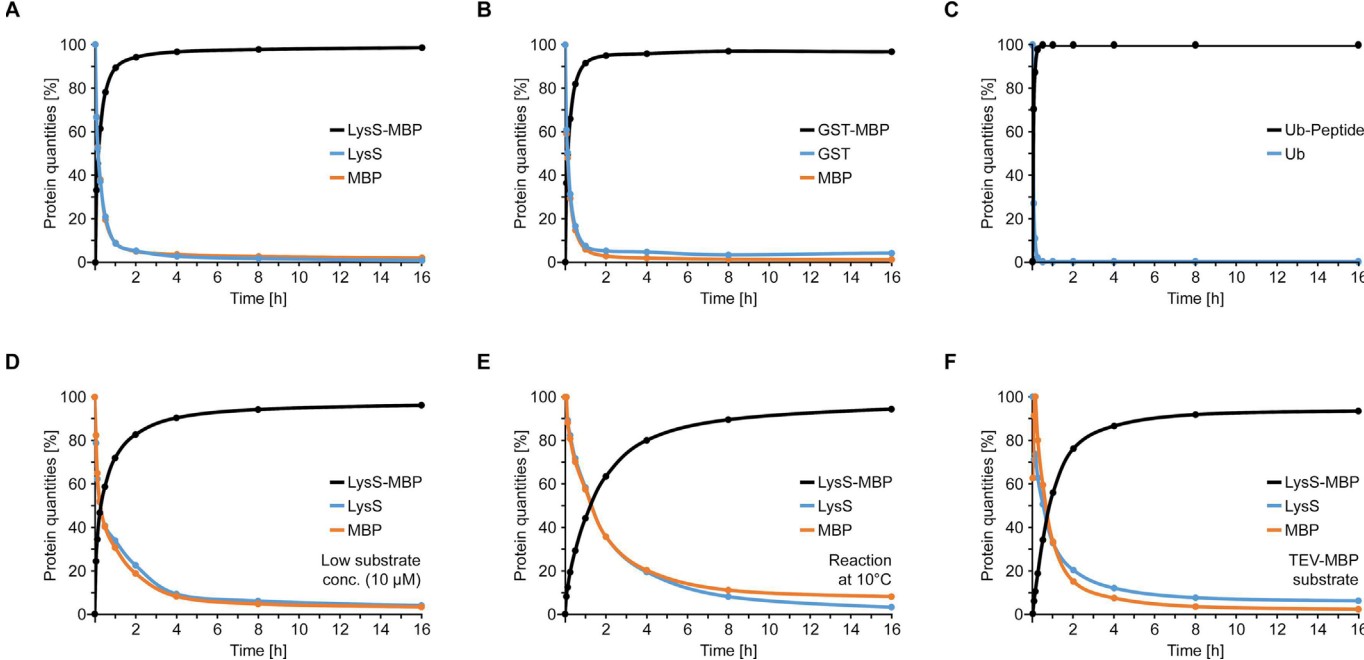

**Appendix 1—figure 1.** Visualization of the educt and product quantities in *Figure 3A–F*.

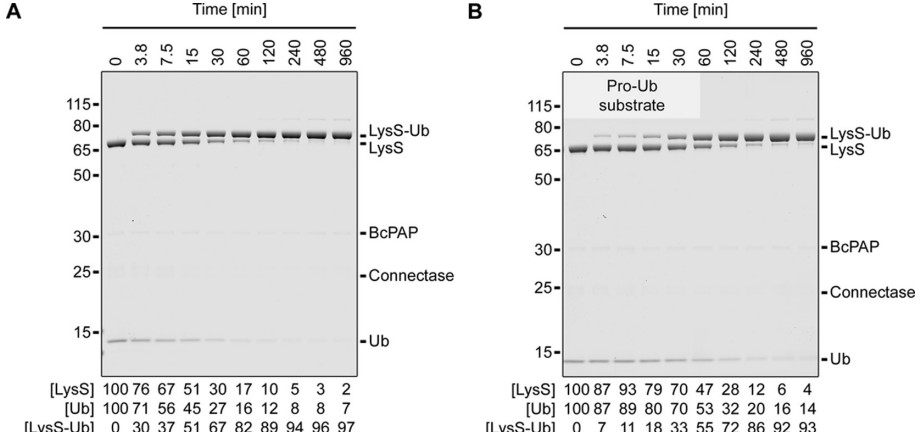

**Appendix 1—figure 2.** Complete protein-protein ligations. Shown are SDS-PAGE time course analyses of ligation reactions using Lysine-tRNA ligase (LysS; C-terminal ELASKDPGAFDADPLVVEI sequence) and Ubiquitin (Ub; N-terminal AGAFDADPLVVEI (**A**) or PAGAFDADPLVVEI (**B**) sequence) as substrates. Both reactions were performed with 1 eq. substrates (100 µM), 0.033 eq. Connectase, and 0.066 eq. BcPAP at 22°C. A densitometric analysis of the protein bands is shown below each experiment. For the substrates, the values reflect the substrate band density relative to the substrate band in the control sample (0 min); for the products, the values reflect the product band density relative to the total band density (substrates + products).

The online version of this article includes the following source data for appendix 1—figure 2:

**Appendix 1—figure 2—source data 1.** PDF file containing original gel images, indicating relevant bands.

**Appendix 1—figure 2—source data 2.** Original gel images.

**Appendix 1—figure 2—source data 3.** Measurements and Calculations.

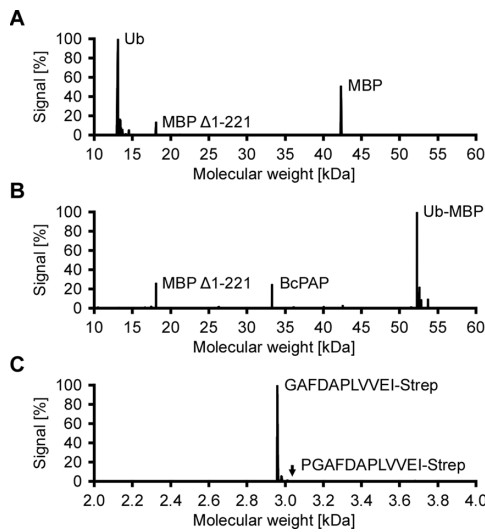

**Appendix 1—figure 3.** LC-MS analysis of an equimolar Ub-MBP mixture before (**A**) and after (**B, C**) conjugation. A mixture of 100 µM Ub-ELASKDPGAFDADPLVVEI-Strep and 100 µM AGAFDADPLVVEI-MBP was analyzed before (**A**) and after (**B**) incubation with 0.033 eq. Connectase and 0.066 eq. BcPAP. The reaction byproduct GAFDADPLVVEI-Strep (**C**) appeared as an extra peak upon conjugation. The signal intensities in the plots were normalized to the most intense peak. MBP was detected both as a full-length version and as an N-terminally truncated version (MBP Δ1–221) without Connectase recognition sequence. The truncated version was not detected by SDS-PAGE (*Figure 3*), suggesting a low abundance in the sample.

The online version of this article includes the following source data for appendix 1—figure 3:

**Appendix 1—figure 3—source data 1.** Measurements and Calculations.

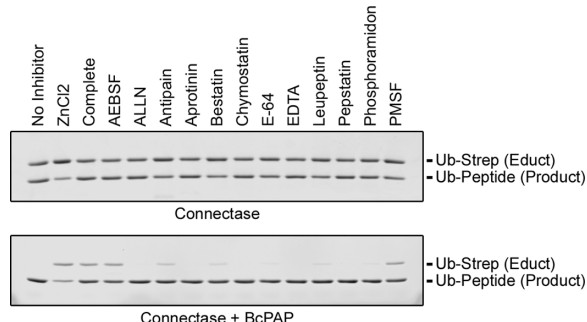

**Appendix 1—figure 4.** Effect of protease inhibitors on BcPAP activity. Shown is the conjugation of Ub-Strep (educt, 1 eq.) to AGAFDADPLVVEI peptide (1 eq.) in presence of different protease inhibitors. The reaction catalyzed by Connectase (upper gel) is not inhibited by these substances and results in an equilibrium between Ub-Strep educt and Ub-Peptide product (as in *Figure 2*). In a reaction with Connectase and BcPAP (lower gel), up to 100% Ub-peptide product is formed. Lower product yields indicate an inhibition of BcPAP. This effect is most pronounced for the serine protease inhibitors AEBSF, PMSF, and a commercial AEBSF-containing inhibitor mix ("complete"). ZnCl$_2$, which had been reported previously as a BcPAP inhibitor[24], led to the precipitation of Connectase and BcPAP.

The online version of this article includes the following source data for appendix 1—figure 4:

**Appendix 1—figure 4—source data 1.** PDF file containing original gel images, indicating relevant bands.

**Appendix 1—figure 4—source data 2.** Original gel images.

Biochemistry and Chemical Biology

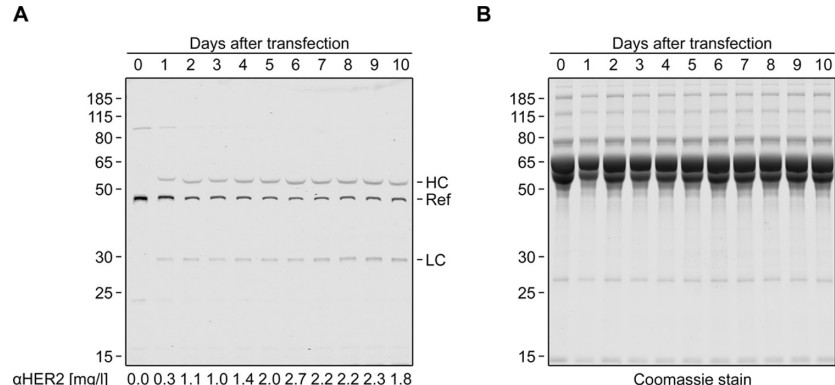

**Appendix 1—figure 5.** Quantification of αHER2 antibodies in cell culture medium. Heavy (HC) and light (LC) antibody chains with a C-terminal Connectase recognition sequence were expressed in HEK293 cells. The medium with the exported antibodies was exchanged daily, allowing the monitoring of antibody expression levels by in-gel fluorescence (**A**). In this western blot alternative, Connectase is used to fuse fluorophores to the target proteins (HC, LC) and a reference protein (Ref). By comparing the intensity of the resulting fluorescent bands, daily antibody expression levels can be estimated (lower panel, see methods). A Coomassie stain of the same gel (**B**) shows all proteins in the cell culture medium samples. The experiment is described and discussed in detail in a previously published paper *Fuchs, 2024*. It is also depicted here because it shows the production of antibodies used in *Figure 4* and highlights the use of the Connectase recognition sequence on a protein of interest for different applications: protein detection and quantification (this figure), and protein conjugation (*Figure 4*). This figure was originally published in Fuchs, A.C.D. Detection and quantification of C-terminally tagged proteins by in-gel fluorescence. *Sci Rep* **14**, 15697 (2024); https://doi.org/10.1038/s41598-024-66132-8. It is licensed under a Creative Commons Attribution 4.0 International License (https://creativecommons.org/licenses/by/4.0/). It was created by the author. Compared to the original image, the signal ratio text line was removed.

The online version of this article includes the following source data for appendix 1—figure 5:

**Appendix 1—figure 5—source data 1.** PDF file containing original gel images, indicating relevant bands.

**Appendix 1—figure 5—source data 2.** Original gel images.

**Appendix 1—figure 5—source data 3.** Measurements and Calculations.

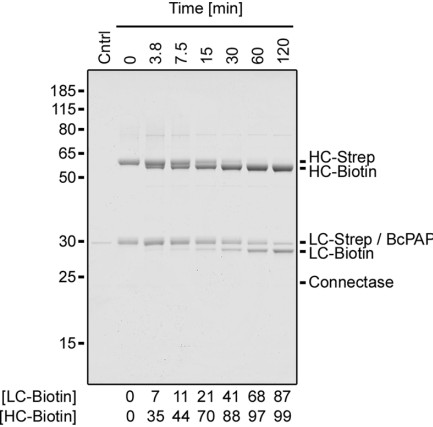

**Appendix 1—figure 6.** Time course of an antibody biotinylation reaction. The heavy chain (HC) and light chain (LC) of αHER2 (human epidermal growth factor receptor 2) antibodies were produced with a C-terminal Connectase recognition sequence and a Streptavidin tag (HC-Strep, LC-Strep). In the reactions (25 µM αHER2, 100 µM subunits), 0.033 eq. Connectase, 0.066 eq. BcPAP, 22°C, the Streptavidin tag is replaced by a shorter biotinylated peptide (1 eq.). The figure shows an SDS-PAGE analysis of the reaction time course. Below the gel, a densitometric quantification of product bands relative to educt bands is presented. For this calculation, the BcPAP density from the control lane (Cntrl) was subtracted from the combined LC-Strep/BcPAP band.

The online version of this article includes the following source data for appendix 1—figure 6:

**Appendix 1—figure 6—source data 1.** PDF file containing original gel images, indicating relevant bands.

**Appendix 1—figure 6—source data 2.** Original gel images.

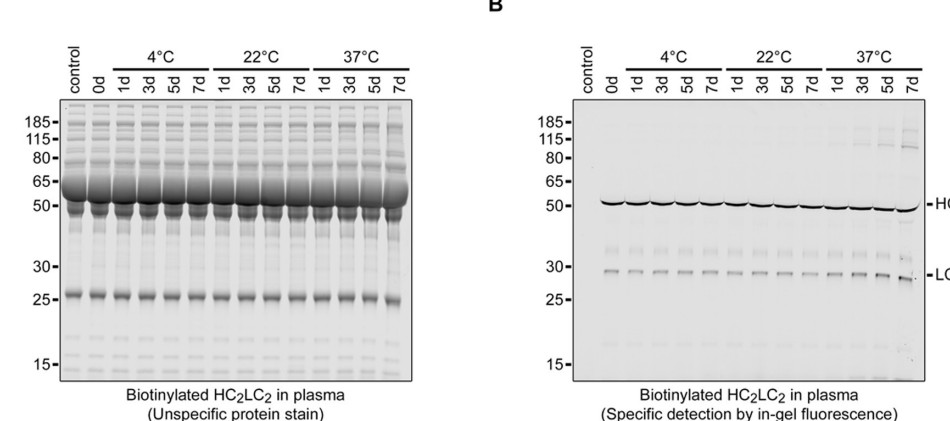

**Appendix 1—figure 7.** Stability of antibody conjugates in human plasma. Antibodies containing C-terminal Connectase recognition sequences were biotinylated (1 μM $HC_2LC_2$-$Biotin_4$) and mixed with human plasma (containing ~86 μM unconjugated $HC_2LC_2$). The mixture was incubated for up to 7 days at 4°C, 22°C, or 37°C, then centrifuged, and the supernatant analyzed by SDS-PAGE. (**A**) Coomassie G-250 staining provides a nonspecific detection of all proteins. (**B**) In-gel fluorescence (*Fuchs, 2023*, *Fuchs, 2024*) specifically detects proteins with Connectase recognition sequences, i.e., HC-Biotin and LC-Biotin, but shows no bands for the plasma control sample without $HC_2LC_2$-$Biotin_4$ (first lane). Throughout the time course, both the HC and LC bands remain visible with consistent intensity, indicating that the biotinylated Connectase recognition sequence (ELASKDAGAFDADPLVVEIK-Biotin) remains intact under all tested conditions.

The online version of this article includes the following source data for appendix 1—figure 7:

**Appendix 1—figure 7—source data 1.** PDF file containing original gel images, indicating relevant bands.

**Appendix 1—figure 7—source data 2.** Original gel images.

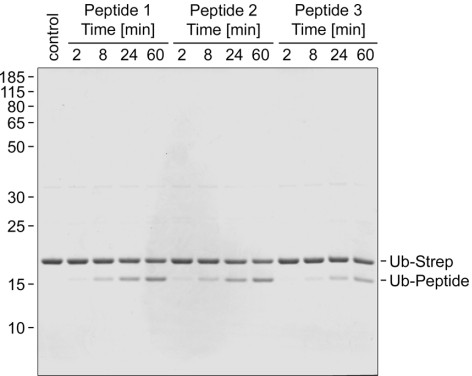

**Appendix 1—figure 8.** Determination of the minimal Connectase recognition sequence. Connectase acts on a linker sequence derived from its physiological interaction partner, Methyltransferase A (MtrA). In an initial characterization (*Fuchs et al., 2021*), this sequence was identified as RELASKDPGAFDADPLVVEI. It remained unclear, whether it could be further shortened from the N-terminal side. The depicted gel shows the Connectase-mediated conjugation of Ub-Strep to RELASKDPGAFDADPLVVEI (peptide 1), ELASKDPGFDADPLVVEI (peptide 2), or LASKDPGAFDADPLVVEI (peptide 3). The product, Ub-peptide, is formed at a similar rate with peptide 1 (relative ligation rate determined by densitometric analysis: 94%) and peptide 2 (100%), but at a reduced rate when using peptide 3 (47%). This suggests that ELASKDPGAFDADPLVVEI is sufficient for efficient conjugation reactions. The protein substrates employed in this paper have the C-terminal RELASKDPGAFDADPLVVEI sequence, with the additional N-terminal arginine serving as a small linker.

The online version of this article includes the following source data for appendix 1—figure 8:

**Appendix 1—figure 8—source data 1.** PDF file containing original gel images, indicating relevant bands.

**Appendix 1—figure 8—source data 2.** Original gel images.

**Appendix 1—figure 8—source data 3.** Measurements and Calculations.

