## [Editor Report · eLife Assessment]

This revision of **important** work is a versatile addition to the chemical protein modifications and bioconjugation toolbox in synthetic biology. The technology developed cleverly uses Connectase to irreversibly fuse proteins of interest together so they can be studied in their native context, with **compelling** well-controlled data showing the technique works for various protein partners. This work will help multiple fields to explore multi-function constructs in basic synthetic biology. This work will also be of interest to those studying fusion oncoproteins commonly expressed in various human pathologies.

---

## [Referee Report · Reviewer #1 (Public review)]

Fuchs describes a novel method of enzymatic protein-protein conjugation using the enzyme Connectase. The author is able to make this process irreversible by screening different Connectase recognition sites to find an alternative sequence that is also accepted by the enzyme. They are then able to selectively render the byproduct of the reaction inactive, preventing the reverse reaction, and add the desired conjugate with the alternative recognition sequence to achieve near-complete conversion. I agree with the authors that this novel enzymatic protein fusion method has several applications in the field of bioconjugation, ranging from biophysical assay conduction to therapeutic development. Previously the author has published on the discovery of the Connectase enzymes and has shown its utility in tagging proteins and detecting them by in-gel fluorescence. They now extend their work to include the application of Connectase in creating protein-protein fusions, antibody-protein conjugates, and cyclic/polymerized proteins. As mentioned by the author, enzymatic protein conjugation methods can provide several benefits over other non-specific and click chemistry labeling methods. Connectase specifically can provide some benefits over the more widely used Sortase, depending on the nature of the species that is desired to be conjugated. Overall, this method provides a novel, reproducible way to enzymatically create protein-protein conjugates.

The manuscript is well-written and will be of interest to those who are specifically working on chemical protein modifications and bioconjugation.

Comments on revisions:

The authors have improved the manuscript significantly by clarifying the questions raised adding new text, providing additional references and/or adding additional data. The thorough study and efficiency of the method for enzymatic protein-protein conjugation using the enzyme Connectase warrants publication of this manuscript in its current form.

---

## [Referee Report · Reviewer #2 (Public review)]

Summary:

Unlike previous traditional protein fusion protocols, the author claims their proposed new method is fast, simple, specific, reversible, and results in a complete 1:1 fusion. A multi-disciplinary approach from cloning and purification, biochemical analyses, and proteomic mass spec confirmation revealed fusion products were achieved.

Strengths:

The author provides convincing evidence that an alternative to traditional protein fusion synthesis is more efficient with 100% yields using connectase. The author optimized the protocol's efficiency with assays replacing a single amino acid and identification of a proline aminopeptidase, Bacilius coagulans (BcPAP), as a usable enzyme to use in the fusion reaction. Multiple examples including Ubiquitin, GST, and antibody fusion/conjugations reveal how this method can be applied to a diverse range of biological processes.

Weaknesses:

Though the ~100% ligation efficiency is an advancement, the long recognition linker may be the biggest drawback. For large native proteins that are challenging/cannot be synthesized and require multiple connectase ligation reactions to yield a complete continuous product, the multiple interruptions with long linkers will likely interfere with protein folding, resulting in non-native protein structures. This method will be a good alternative to traditional approaches as the author mentioned but limited to generating epitope/peptide/protein tagged proteins, and not for synthetic protein biology aimed at examining native/endogenous protein function in vitro.

---

## [Author Response]

The following is the authors’ response to the original reviews.

**Reviewer #1 (Public review):**
Fuchs describes a novel method of enzymatic protein-protein conjugation using the enzyme Connectase. The author is able to make this process irreversible by screening different Connectase recognition sites to find an alternative sequence that is also accepted by the enzyme. They are then able to selectively render the byproduct of the reaction inactive, preventing the reverse reaction, and add the desired conjugate with the alternative recognition sequence to achieve near-complete conversion. I agree with the authors that this novel enzymatic protein fusion method has several applications in the field of bioconjugation, ranging from biophysical assay conduction to therapeutic development. Previously the author has published on the discovery of the Connectase enzymes and has shown its utility in tagging proteins and detecting them by in-gel fluorescence. They now extend their work to include the application of Connectase in creating protein-protein fusions, antibody-protein conjugates, and cyclic/polymerized proteins. As mentioned by the author, enzymatic protein conjugation methods can provide several benefits over other non-specific and click chemistry labeling methods. Connectase specifically can provide some benefits over the more widely used Sortase, depending on the nature of the species that is desired to be conjugated. However, due to a similar lengthy sequence between conjugation partners, the method described in this paper does not provide clear benefits over the existing SpyTag-SpyCatcher conjugation system. Additionally, specific disadvantages of the method described are not thoroughly investigated, such as difficulty in purifying and separating the desired product from the multiple proteins used. Overall, this method provides a novel, reproducible way to enzymatically create protein-protein conjugates.The manuscript is well-written and will be of interest to those who are specifically working on chemical protein modifications and bioconjugation.

I'd like to comment on two points.

(1) The benefits over the SpyTag-SpyCatcher system. Here, the conjugation partners are fused via the 12.3 kDa SpyCatcher protein, which is considerably larger than the Connectase fusion sequence (19 aa). This is mentioned in the introduction (p. 1 ln 24-26). Furthermore, SpyTag-SpyCatcher fusions are truly irreversible, while Connectase/BcPAP fusions may be reversed (p. 8, ln 265-273). For example, target proteins (e.g., AGAFDADPLVVEI-Protein) may be covalently fused to functionalized magnetic beads (e.g., Bead-ELASKDPGAFDADPLVVEI) in order to perform a pulldown assay. After the assay, the target protein and any bound interactors could be released from the beads by the addition of a Connectase / peptide (AGAFDAPLVVEI) mixture.

In a related technology, the SpyTag-SpyCatcher system was split into three components, SpyLigase, SpyTag and KTag (Fierer et al., PNAS 2014). The resulting method introduces a sequence between the fusion partners (SpyTag (13aa) + KTag (10aa)), which is similar in length to the Connectase fusion sequence (p. 8, ln 297 - 298). Compared to the original method, however, this approach seems to require longer incubation times, while yielding less fusion product (Fierer et al., Figure 2).

(2) Purification of the fusion product. The method is actually advantageous in this respect, as described in the discussion (p. 8, ln 258-264). Examples are now provided in Figure 6.

**Reviewer #2 (Public review):**
Summary:Unlike previous traditional protein fusion protocols, the author claims their proposed new method is fast, simple, specific, reversible, and results in a complete 1:1 fusion. A multi-disciplinary approach from cloning and purification, biochemical analyses, and proteomic mass spec confirmation revealed fusion products were achieved.Strengths:The author provides convincing evidence that an alternative to traditional protein fusion synthesis is more efficient with 100% yields using connectase. The author optimized the protocol's efficiency with assays replacing a single amino acid and identification of a proline aminopeptidase, Bacilius coagulans (BcPAP), as a usable enzyme to use in the fusion reaction. Multiple examples including Ubiquitin, GST, and antibody fusion/conjugations reveal how this method can be applied to a diverse range of biological processes.Weaknesses:Though the ~100% ligation efficiency is an advancement, the long recognition linker may be the biggest drawback. For large native proteins that are challenging/cannot be synthesized and require multiple connectase ligation reactions to yield a complete continuous product, the multiple interruptions with long linkers will likely interfere with protein folding, resulting in non-native protein structures. This method will be a good alternative to traditional approaches as the author mentioned but limited to generating epitope/peptide/protein tagged proteins, and not for synthetic protein biology aimed at examining native/endogenous protein function in vitro.

The assessment is fair, and I have no further comments to add.

**Reviewer #1 (Recommendations for the authors):**
Major/Experimental Suggestions:(1) Throughout the paper only one reaction shown via gels had 100% conversion to desired product (Figure 3C). It is misleading to title a paper with absolutes such as "100% product yield", when the majority of reactions show >95% product yield, without any purification. Please change the title of the manuscript to something along the lines of "Novel Irreversible Enzymatic Protein Fusions with Near-Complete Product Yield".

The conjugation reaction is thermodynamically favored. It is driven by the hydrolysis of a peptide bond (P|GADFDADPLVVEI), which typically releases 8 - 16 kJ/mol energy. This should result in a >99.99% complete reaction (DG° = -RT ln (Product/Educt)). In line with this, 99% - 100% of the less abundant educts (LysS, Figure 3A; MBP, Figure 3B; Ub-Strep, Figure 3C) are converted in the time courses (Figure 3D-F show different reaction conditions, which slow down conjugate formation). 100% conversion are also shown in Figure 5, Figure 6, and Figure S4. Likewise, 99.6% relative fusion product signal intensity in an LCMS analysis (Figure S2) after 4h reaction time (0.13% and 0.25% educts). In this experiment, the proline had been removed from 99.8% of the peptide byproducts (P|GADFDADPLVVEI). It is clear that this reaction is still ongoing and that >99.99% of the prolines will be removed from the peptides in time. These findings suggest that the conjugation reaction gradually slows down the less educt is available, but eventually reaches completion.

For some experiments, lower product yields (e.g. 97% in Figure 3B) are reported in the paper. These were calculated with Yield = 100% x Product / (Educt1 + Educt 2 + Product). With this formula, 100% conjugation can only be achieved with exactly equimolar educt quantities, because both educt 1 and educt 2 need to be converted entirely. If one educt 1 is available in excess, for example because of protein concentration measurement inaccuracies or pipetting errors, some of it will be left without fusion partner. In case of Figure 3B, 3% more GST seemed to have been in the mixture. These are methodological inaccuracies.

(2) Please provide at least one example of a purified desired product, and mention the difficulties involved as a disadvantage to this particular method. Separating BcPAP, Connectase, and the desired protein-protein conjugate may prove to be quite difficult, especially when Connectase cleaves off affinity tags.

Examples are now provided in Figure 6. As described in the discussion (p. 8, ln 258-264), the simple product purification is one of the advantages of the method.

(3) For the antibody conjugate, please provide an example of conjugating an edduct that would prove to be more useful in the context of antibodies. For example, as you mention in the introduction, conjugation of fluorophores, immobilization tags such as biotin, and small molecule linker/drugs are useful bioconjugates to antibodies.

Antibody-biotinylation is now shown in Figure S6; Antibody-fluorophore conjugates are part of Figures S5 and S7.

(4) Please assess the stability of these protein-protein conjugates under various conditions (temperature, pH, time) to ensure that the ligation via Connectase is stable over a broad array of conditions. In particular, a relevant antibody-conjugate stability assay should be done over the period of 1-week in both buffer and plasma to show applicability for potential therapeutics.

The stability of an antibody-biotin conjugate in blood plasma over 7 days at different temperatures is now shown in Figure S7.

Generally, Connectase introduces a regular peptide bond (Asp-Ala) with a high chemical and physical stability (e.g. 10 min incubation at 95°C in SDS-PAGE loading buffer; H2O-formic acid / acetonitrile gradients for LC-MS). The sequence may be susceptible to proteases, although this is not the case in HEK293 cells (antibody expression), *E. coli*, or blood plasma (Figure S7).

(5) Please conduct functional assays with the antibody-protein/peptide conjugates to show that the antibody retains binding capabilities to the HER-2 antigen and the modification was site-selective, not interfering with the binding paratope or binding ability of the antibody in any way. This can be done through bio-layer interferometry, surface plasmon resonance, ELISA, etc.

We plan the immobilization of the HER2 antibody on microplates and its use in an ELISA. However, this experiment requires significant testing and optimizations. It will be part of a future paper on the use of Connectase for protein immobilization.

For now, the mass spectrometry data provide clear evidence of a single site-selective conjugation, as the C-terminal ELASKDPGAFDADPLVVEI-Strep sequence is replaced by ELASKDAGAFDADPLVVEI(-Ub). Given that the conjugation sites at the C-termini are far from the antigen binding sites, and have already been used in a number of other approaches (e.g., SpyTag, SnapTag, Sortase), it appears unlikely that these conjugations interfere with antigen binding.

(6) Please include gels of all proteins used in ligation reactions after purification steps in the SI to show that each species was pure.

The pure proteins are now shown in Figure S9.

(7) Please provide the figures (not just tables) of LC/MS deconvoluted mass spectra graphs for all conjugates, either in the main text or the SI.

Please specify which spectra you are missing. I believe all relevant spectra are shown in Figures 4, 5, and S3. The primary data can be found in Dataset S2.

(8) Please provide more information in the methods section on exactly how the densitometry quantification of gel bands was performed with ImageJ.

Details on the quantification with Image Studio Lite 5.2 were added in the method section (p. 17, ln 461-463).

Minor Suggestions:(1) Page 1, line 19: can include one sentence on what assays these particular bioconjugations are usefule for (e.g. internalization cell studies, binding assays, etc.)

I prefer not to provide additional details here to keep the text concise and focused.

(2) Page 1, line 22: "three to ten equivalents" instead of 3x-10x.

Done.

(3) Page 1, line 23: While NHS labeling is widely considered non-specific, maleimide conjugation to free cysteines is generally considered specific for engineered free cysteine residues, since native proteins often do not have free cysteine residues available for conjugation. If you are referring to the potential of maleimides to label lysines as well, that should be specifically stated.

I modified the sentence, now stating that these methods are "can be" unspecific.

As pointed out, it is possible to achieve specificity by eliminating all other free cysteines and/or engineering a cysteine in an appropriate position. In many other cases, however (e.g., natural antibodies), several cysteines are available, or the sample contains other proteins/peptides. I did not want to go into more detail here and refer to the cited review.

(4) Page 1, line 31: "and an oligoglycine G(1-5)-B"

Done.

(5) Page 1, line 34: It is not clear where in the source these specific Km values are coming from, considering these are variable based on specific conditions/substrates and tend to be reaction-specific.

I cited another review, which lists the same values, along with a few other measurements (Jacobitz et al., Adv Protein Chem Struct Biol 2017, Table 2). It is clear that each of these measurements differs somewhat, but they are generally comparable (K_M_(LPETG) = 5500 - 8760 µM; K_M_(GGGGG) = 140 - 196 µM). I chose the cited study (Frankel et al., Biochemistry 2005), because it also investigated hydrolysis rates. In this study, the measurements are derived from the plots in Figure 2.

(6) Page 1, line 47: the comparison to western blots feels a little like apples to oranges, even though this comparison was made in previous literature. Engineering an expressed protein to have this tag and then using the tag to detect and quantify it, feels more akin to a tagging/pull down assay than a western blot in which unmodified proteins are easily detected.

It is akin to a frequently used type of western blots with tag-specific antiboies, e.g. Anti-His_6_, -Streptavidin, -His_6_, -HA ,-cMyc, -Flag. I modified the sentence to clarify this.

(7) Page 2, line 51: "Connectase cleaves between the first D and P amino acids in the recognition sequence, resulting in an N-terminal A-ELASKD-Connectase intermediate and a C-terminal PGAFDADPLVVEI peptide."

I prefer the current sentence, because we assume that a bond between the aspartate and Connectase is formed before PGAFDADPLVVEI is cleaved off.

(8) Page 3, line 94: "Exact determination is not possible due to reversibility of the reaction", the way it is stated now sounds like it is a flaw in the methods. Also, update Figure 2 to read "Estimated relative ligation rate".

Done.

(9) Page 3, lines 101-107: This is worded in a confusing way. It can either be X_1_ or X_2_ that is inactivated depending on if the altered amino acid is on the original protein sequence or on the desired edduct to conjugate. You first give examples of how to render other amino acids inactive, but then ultimately state that proline made inactive, so separate the two distinct possibilities a bit more clearly.

The reaction requires the inactivation of X_1_, without affecting X_2_ (ln 100 - 102). This is true, no matter whether it is X_1_ = A, C, S, or P that is inactivated. I added a sentence to clarify this (ln 102 – 103).

(10) Page 4, line 118: Give a one-sentence justification for why these proteins were chosen to work with (easy to express, stable, etc).

Done.

(11) Page 5, line 167: "payload molecules".

Done.

(12) Page 5, lines 170-173: Word this more clearly- "full conversion with many of these methods is difficult on antibodies due to each heavy and light chain being modified separately, resulting in only a total yield of 66% DAR4 even when 90% of each chain is conjugated."

I rephrased the section.

(13) Page 8, line 290: Discuss other disadvantages of this method including difficulties purifying and in incorporating such a long sequence into proteins of interest.

Product purification is shown in the new Figure 6. As stated above, I consider the simple purification process an advantage of the method. The genetic incorporation of the sequence into proteins is a routine process and should not make any difficulties. The disadvantages of long linker sequences between fusion partners are now discussed (p.8 – 9, ln 300-302).

(14) Page 10, line 341: 'The experiment is described and discussed in detail in a previously published paper.31"

Done.

**Reviewer #2 (Recommendations for the authors):**
Minor Points:(1) It's unclear how the author derived 100% ligation rate with X = Proline in Figure 2 when there is still residual unligated UB-Strep at 96h. Please provide an expanded explanation for those not familiar with the protocol. Is the assumption made that there will be no UB-Strep if the assay was carried out beyond 96h?

I clarified the figure legend. The assay shows the formation of an equilibrium between educts and products. Therefore, only ~50% Ub-Strep is used with X = Proline (see p. 2, ln 79 - 81). The "relative ligation rate" refers to the relative speed with which this equilibrium is established. The highest rate is seen with X = Proline, and it is set to 100%. The other rates are given relative to the product formation with X = Proline.

(2) Though the qualitative depiction of the data in Figure 3 is appreciated, an accompanying graphical representation of the data in the same figure will greatly enhance reception and better comprehension of several of the author's conclusions.

Graphs are now shown in Figure S1.

(3) Figure 3 panel E is misaligned. Please align it with panel B above it.

Done, thank you.

(4) The author refers to 'The resulting circular assemblies (37% UB2...)' in the text but identifies it as UB-C2 in Figure 5B. Is this a mistake or does UB2 refer to another assembly not mentioned in the Figures? Please check for inconsistencies.

All circular assemblies are now labeled Ub-C _1-6_.

(5) Finishing with a graphical schematic that depicts the entire protocol in a simple image would be much appreciated and well-received by readers. Including the scheme with A and B proteins, the recognition linkers, the addition of connectase and BcPAP, etc. to the final resulting protein with connected linker.

A graphical summary of the reaction is now included in Figure 6.